# Unified 3D Segmenter As Prototypical Classifiers

**Zheyun Qin[1]\*, Cheng Han[2]\*, Qifan Wang[3], Nie Xiushan[4], Yilong Yin[1]†, Xiankai Lu[1]†**

[1]Shandong University, [2]Rochester Institute of Technology, [3]Meta AI, [4]Shandong Jianzhu University

## Abstract

The task of point cloud segmentation, comprising semantic, instance, and panoptic segmentation, has been mainly tackled by designing task-specific network architectures, which often lack the flexibility to generalize across tasks, thus resulting in a fragmented research landscape. In this paper, we introduce PROTOSEG, a prototype-based model that unifies semantic, instance, and panoptic segmentation tasks. Our approach treats these three homogeneous tasks as a classification problem with different levels of granularity. By leveraging a Transformer architecture, we extract point embeddings to optimize prototype-class distances and dynamically learn class prototypes to accommodate the end tasks. Our prototypical design enjoys simplicity and transparency, powerful representational learning, and ad-hoc explainability. Empirical results demonstrate that PROTOSEG outperforms concurrent well-known specialized architectures on 3D point cloud benchmarks, achieving 72.3%, 76.4% and 74.2% mIoU for semantic segmentation on S3DIS, ScanNet V2 and SemanticKITTI, 66.8% mCov and 51.2% mAP for instance segmentation on S3DIS and ScanNet V2, 62.4% PQ for panoptic segmentation on SemanticKITTI, validating the strength of our concept and the effectiveness of our algorithm. The code and models are available at `https://github.com/zyqin19/PROTOSEG`.

## 1 Introduction

Point cloud segmentation entails partitioning a collection of 3D data points into meaningful categories to interpret complex 3D scenes. Three classical subdivision segmentation tasks are involved: semantic segmentation, which assigns a class label to each point based on its class; instance segmentation, which distinguishes between individual instances of the same class; and panoptic segmentation, which combines semantic and instance segmentation to address amorphous or uncountable regions. These tasks use distinct technical approaches, which have advanced each individual task but lack flexibility to generalize to other tasks. This methodological convention leads to fragmented research efforts.

For the purpose of advancing the field of point cloud segmentation in a synergistic manner, a shift from task-specific network architectures towards a more universal framework is imperative. It is natural to question why the existing methods cannot achieve such unification, and we address this issue in § 2 by examining the relationships and distinctions between current approaches. Many of these approaches are customized for individual tasks, resulting in data representation, feature extraction, and prediction head inconsistencies that hinder integration into a unified framework [1]. Furthermore, task-specific architectures tend to overfit their respective tasks, reducing their generalizability when applied to other segmentation tasks, thus hindering the practical unification of segmentation tasks.

In light of the above discussion, our research question becomes more fundamental: what epistemological framework can effectively embrace this shift? In response, we revisit the prototype paradigm and point clouds' natural properties (irregular and sparse), and propose that point cloud segmentation naturally aligns with a classification problem. This foundational perspective shapes our research direction and motivates the creation of our prototype-anchored classification method, allowing us to unify various segmentation tasks within a single framework seamlessly. In § 3, we introduce

---

*Equal contribution. †Corresponding authors.

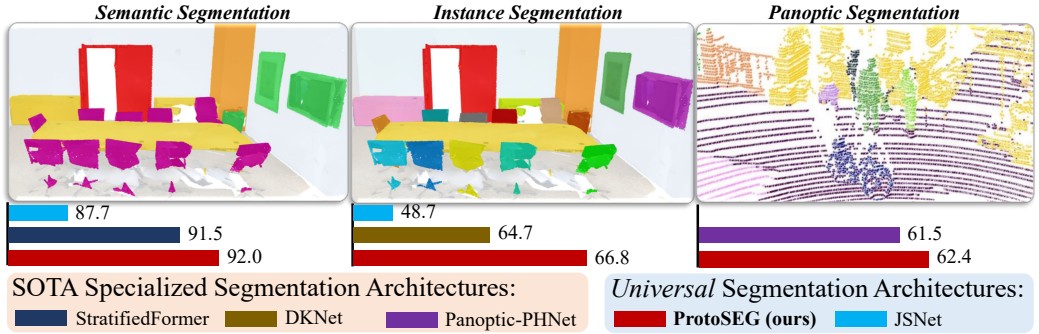

Figure 1: PROTOSEG unifies three 3D segmentation tasks (*i.e.*, semantic, instance, and panoptic) from the prototypical view, and greatly suppresses existing specialized and unified models.

PROTOSEG, a prototype-anchored classification method that unifies semantic, instance, and panoptic segmentation. To achieve this, we employ a Transformer [2] backbone to extract point embeddings and represent classes as prototypes. This approach allows for task-specific outputs based on various granularity annotations (*i.e.*, semantic, instance, and scene), without architectural modification. By utilizing a dynamic association and updating mechanism, PROTOSEG captures class-wise characteristics and intra-class variance. Ultimately, PROTOSEG creates well-structured embedded spaces with inter-class separation and intra-class compactness by optimizing point-prototype-class distances.

Concretely, our innovations focus on three aspects. ① **Simplicity and transparency:** PROTOSEG has an intuitive working mechanism [3] as the statistical meaning of class sub-centroids makes it elegant and easy to understand. This leads to the automatic discovery of underlying data structures, as the latent distribution of each class is automatically mined and fully captured as a set of representative local means (see §3.2). This contrasts with non-prototype methods [4] that learn a single weight (or query) vector per class, which may struggle to accommodate rich intra-class variations [5, 6]. ② **Representation learning:** PROTOSEG achieves point-wise classification through comparisons between data samples and class sub-centroids in the embedding space. This distance-based nature enables PROTOSEG to blend unsupervised sub-pattern mining – class-wise clustering (see Eqs. 13-14) with supervised representation learning – learnable prototype classification (see Eq. 15) in a synergistic manner. Significant local patterns are automatically identified to facilitate classification decisions, while the supervisory signal from classification directly optimizes the representation, in turn boosting meaningful clustering and discriminative features learning. ③ **Ad-hoc explainability:** PROTOSEG is a transparent classifier, imbued with a built-in case-based reasoning process, establishing *IF⋯Then* rules [7]. This is achieved by allowing the class sub-centroids to serve as representative samples from the training set, thus providing human-understandable explanations for each prediction (see Fig. 3). Such an ad-hoc explainability adheres to the internal decision-making process, thereby differentiating PROTOSEG from many existing methods [8] that fail to elucidate precisely how a model works.

Incorporating these innovations, we present an elegant, general, and flexible framework for point cloud segmentation that achieves remarkable results (see Fig. 1) on widely recognized 3D point cloud semantic, instance, and panoptic segmentation tasks. Specifically, our method achieves notable performance gains in terms of $91.5\% \rightarrow 92.2\%$ OAcc, $75.2\% \rightarrow 76.4\%$ mIoU and $75.4\% \rightarrow 76.3\%$ mIoU for semantic segmentation on S3DIS [9] Area 5 and ScanNet V2 [10] *test/val.* sets, $64.7\% \rightarrow 66.8\%$ mCov, $50.6\% \rightarrow 51.2\%$ mAP and $67.6\% \rightarrow 68.4\%$ mAP$_{50}$ for instance segmentation on S3DIS [9] Area 5 and ScanNet V2 [10] *test/val.* sets, and $61.5\% \rightarrow 62.4\%$ PQ for panoptic segmentation on SemanticKITTI [11] *test* set when compared to main competitors. Accompanied by a comprehensive series of ablation studies in §5.4, our extensive experiments evaluations confirm the strength of our concept and the effectiveness of our prototype-anchored classification algorithm.

## 2   Existing Task-Specialized Segmentation Models as Non-Prototype Learners

In this section, we provide an overview of existing methods for point cloud segmentation that are tailored to specific tasks, such as semantic, instance, or panoptic segmentation. We will discuss the key aspects of these methods, including data representation, feature extraction, and projection head.

A key challenge is the inherent customization of segmentation methods for specific tasks. From a broader perspective, it's evident that semantic segmentation methods might not directly apply to instance segmentation tasks and vice versa. For instance, the semantic segmentation method PTv2 [2] lacks modules for instance-specific localization and segmentation. This makes it inadequate for capturing the detailed spatial relationships required for precise instance boundary delineation and separation. In contrast, the instance segmentation method SoftGroup [12] uses grouping mechanisms to link points belonging to the same instance. While such a mechanism is essential for instance segmentation, it could introduce unwarranted complexity and overhead in semantic segmentation scenarios, where distinguishing individual instances isn't the primary goal.

On a finer scale, we observe inconsistencies across data representation, feature extraction, and prediction heads (Eqs. 1,2,3) in these methods. To illustrate, let's first explore the differences between the data representations that make up the initial steps of the segmentation. For a given 3D point cloud scene $\mathcal{M}$ comprised of a set of points, data can be represented in multiple ways, including projection-based [13], voxel-based [14, 15] and point-based [16, 17, 18, 19, 2] representations. Mathematically, the input data representation through function $\rho(\cdot)$ can be expressed as:

$$\boldsymbol{X} = \rho(\mathcal{M}). \tag{1}$$

After the data representation is obtained, an encoder $\phi$ is employed to extract point cloud embeddings:

$$\boldsymbol{E} = \phi(\boldsymbol{X}), \tag{2}$$

where $\phi$ incorporates techniques such as object detection [20, 21], clustering [22, 23] or graph learning [24, 25] to enhance the embedding representation based on the backbone network.

Finally, each point embedding $\boldsymbol{e}_i \in \boldsymbol{E}$ is fed into the projection head for segmentation [26]. Let $\boldsymbol{V} = [\boldsymbol{v}^1, \cdots, \boldsymbol{v}^C] \in \mathbb{R}^{C \times D}$ be a set of class-specific vectors, where $C$ is the number of class, $D$ is vector dimension, $\boldsymbol{v}^c \in \mathbb{R}^D$ represents the linear weight or query vector in MLP-based [18, 2, 19, 2] or Query-based models [27, 28] for the $c^{th}$ class. The probability that a point example $\boldsymbol{x}_i \in \boldsymbol{X}$ with embedding $\boldsymbol{e}_i \in \mathbb{R}^D$ is assigned to class $c$ can be expressed with inner product as follows:

$$p(c|\boldsymbol{x}_i) = \frac{\exp((\boldsymbol{v}^c)^\top \boldsymbol{e}_i)}{\sum_{c'=1}^{C} \exp((\boldsymbol{v}^{c'})^\top \boldsymbol{e}_i)}. \tag{3}$$

These above inconsistencies in data representation, feature extraction, and projection head further complicate the unification of segmentation tasks, as the learned features may not be directly transferable across different tasks. In addition, the segmentation methods employed may not be optimal for specific tasks, limiting their applicability and performance in a unified framework. Despite these methods necessitating different processing techniques, thereby complicating the unification of the tasks, they can collectively be categorized as **Non-Prototype Learners**.

## 3 Universal Segmentation Models as Prototype Classifiers

### 3.1 Problem Formulation

To address the challenges, we propose a unified framework that interprets different segmentation tasks as unique granularity classification problems. This approach crafts a prototype classifier, eliminating extra computational load and task-specific architectures. Specifically, let $\mathcal{M}$ be a 3D point cloud scene containing a set of points $\{\boldsymbol{x}_i\}$. The objective of point cloud segmentation is to assign each point $\boldsymbol{x}_i$ to the class assignment set:

$$\textbf{PointSeg}(\mathcal{M}) : \boldsymbol{x}_i \mapsto \{\mathcal{A}_c(\boldsymbol{x}_i) \mid \mathcal{A}_c(\boldsymbol{x}_i) \in \{0, 1\}, c = \{1, \ldots, C\}\} \tag{4}$$

where $\mathcal{A}_c(\boldsymbol{x}_i)$ indicates whether the point $\boldsymbol{x}_i \in \mathcal{M}$ belongs (1) or does not belong (0) to class $c$. Distinct from Eqs. 1-3 in §2, we seamlessly integrate 3D point cloud segmentation under Eq.4, adopting a perspective of prototype-anchored classification, considering various levels of granularity.

### 3.2 Prototypical Classifiers for Segmentation

We propose a unified prototypical classifier for specific segmentation tasks. To achieve this, we leverage a set of prototypes $\boldsymbol{P} = \{\boldsymbol{p}_k^c\}_{c,k=1}^{C,K} \in \mathbb{R}^{D \times C \times K}$, where $\boldsymbol{p}_k^c \in \mathbb{R}^D$ denotes the center of the $k^{th}$ sub-cluster (property) of training point samples belonging to class $c$. For a given point sample $\boldsymbol{x}_i$,

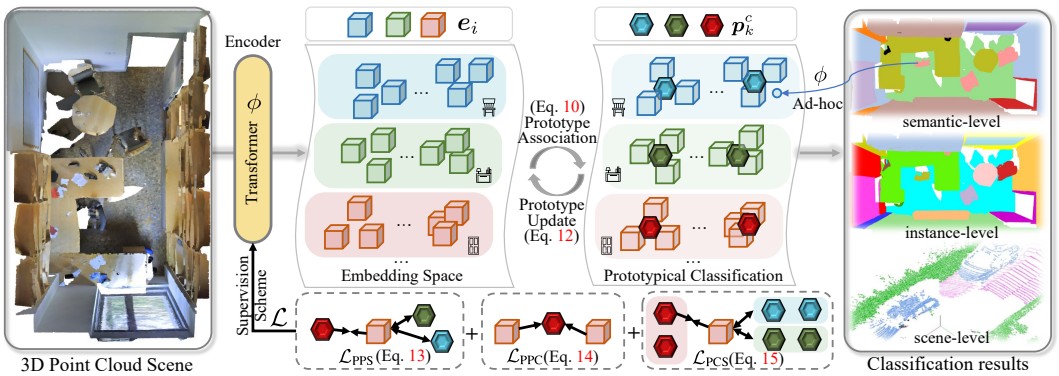

Figure 2: Overall pipeline of PROTOSEG. The classification of points is predicted by evaluating the minimum distance between the embedding $e_i$ (extracted by encoder $\phi$) and prototype $p_k^c$ via our Prototype Association and Update mechanism (see Eq. 10, Eq. 12 in §3.2). Subsequently, supervision scheme optimizes $\phi$ by minimizing the point-prototype-class distance (see Eqs. 13-15 in §3.3). Ad-hoc explains the decision-making process of our classification model, detailed in §5.4.

we make predictions by comparing point embedding $e_i \in \mathbb{R}^D$, extracted using a simple Transformer encoder $\phi$, with prototype $p_k^c$ and assigning the corresponding prototype's class as the response:

$$p(c|\boldsymbol{x}_i) = \frac{\exp(-d_{\boldsymbol{e}_i,c})}{\sum_{c'=1}^{C}\exp(-d_{\boldsymbol{e}_i,c'})}, \quad \text{with } d_{\boldsymbol{e}_i,c} = \arg\min\{\langle\boldsymbol{e}_i, \boldsymbol{p}_k^c\rangle\}_{k=1}^{K}, \tag{5}$$

where the negative cosine distance measure $\langle\cdot,\cdot\rangle$ is defined as $\langle\boldsymbol{e}_i, \boldsymbol{p}_k^c\rangle = -\boldsymbol{e}_i^\top \boldsymbol{p}_k^c$.

To get the informative prototype that can represent the properties of the class, we first associate the point embeddings to the prototypes belonging to the same class (*i.e.*, **Prototype Association**) and then dynamically update prototypes according to the assignments (*i.e.*, **Prototype Update**).

**Prototype Association.** Given embeddings of point samples $\boldsymbol{E}^c = \{\boldsymbol{e}_i^c\}_{i=1}^{N_c} \in \mathbb{R}^{D \times N_c}$ in a training batch and prototypes $\boldsymbol{P}^c = \{\boldsymbol{p}_k^c\}_{k=1}^{K} \in \mathbb{R}^{D \times K}$ for class $c$, we aim to maximize the similarities $\mathcal{J}(\boldsymbol{M})$ between the associated point sample embeddings and corresponding class prototypes:

$$\max_{\boldsymbol{M}} \ \mathcal{J}(\boldsymbol{M}) = \text{Tr}((\boldsymbol{M}^c)^\top(\boldsymbol{P}^c)^\top\boldsymbol{E}^c), \tag{6}$$

where $\text{Tr}(\cdot)$ denotes the matrix trace, $\boldsymbol{M}^c = \{\boldsymbol{m}_{k,i}^c\}_{k,i=1}^{K,N_c} \in \{0,1\}^{K \times N_c}$ is a point-to-prototype permutation matrix that denotes the association between point and prototype, $\boldsymbol{m}_{k,i}^c \in \{0,1\}$ denotes the one-hot assignment vector that assigns the point sample $\boldsymbol{x}_i$ to the prototype $k$ of the class $c$.

To accommodate the representation of all classes for $\boldsymbol{M}^c$, we impose two constraints aimed at avoiding a trivial solution in which all point samples are assigned to a single prototype [29]:

$$(\boldsymbol{M}^c)^\top\mathbf{1}_K = \mathbf{1}_{N_c}, \ (\boldsymbol{M}^c)^\top\mathbf{1}_{N_c} = \frac{N_c}{K}\mathbf{1}_K, \tag{7}$$

where $\mathbf{1}_K, \mathbf{1}_{N_c}$ denotes the tensor of all ones of $K$ or $N_c$ dimensions. The former *Unique assignment Constraint* ensures that each point embedding is assigned to one and only one prototype across all classes. The latter *Equipartition Constraint* enforces that each prototype is selected an equal number of times within the batch on average. The implementation of these constraints is of utmost importance as it significantly enhances the representative capability of the prototypes for each class.

For more intuitive, Eq. 6 can be rewritten as estimating the optimal $\boldsymbol{M}^{c,*}$:

$$\boldsymbol{M}^{c,*} = \max_{\boldsymbol{M}} \text{Tr}((\boldsymbol{M}^c)^\top(\boldsymbol{P}^c)^\top\boldsymbol{E}^c). \tag{8}$$

Following [30], we smooth Eq. 8 with an entropic regularization term $h(\boldsymbol{M}^c)$:

$$\boldsymbol{M}^{c,*} = \max_{\boldsymbol{M}^c} \text{Tr}((\boldsymbol{M}^c)^\top(\boldsymbol{P}^c)^\top\boldsymbol{E}^c) + \kappa h(\boldsymbol{M}^c), \quad s.t. \quad \boldsymbol{M}^c \in \mathbb{R}_+^{K \times N}, \tag{9}$$

where $h(\boldsymbol{M}^c) = \sum_{k,i} -\boldsymbol{m}_{k,i}^c \log \boldsymbol{m}_{k,i}^c$ reduces the randomness of association and avoids most point samples associating to one single prototype. We control the association smoothness using a small value $\kappa$ to ensure that each point embedding is assigned to only one prototype.

With the entropic regularization term in Eq. 9, the problem can be solved using the Sinkhorn-Knopp iteration [30] on the renormalization vectors $\boldsymbol{u}^c \in \mathbb{R}^K$ and $\boldsymbol{v}^c \in \mathbb{R}^{N^c}$. After a few iterations, the vectors $\boldsymbol{u}^c$ and $\boldsymbol{v}^c$ converge, and then the permutation matrix $\boldsymbol{M}^*$ can be directly calculated as:

$$\boldsymbol{M}^{c,*} = \mathbf{diag}(\boldsymbol{u}^c) \exp\left(\frac{(\boldsymbol{P}^c)^\top \boldsymbol{E}^c}{\kappa}\right) \mathbf{diag}(\boldsymbol{v}^c), \tag{10}$$

where $\mathbf{diag}(\cdot)$ denotes a diagonal matrix formed by the input vector. The hyper-parameter $\kappa$ balances the convergence speed and stability of Eq. 10 in addition to smoothing the association (Eq. 9). A smaller $\kappa$ can lead to slower convergence but more accurate results, while a larger $\kappa$ does the opposite. We just use $\kappa = 0.05$ following [30] for our experiments, not extensively fine-tuned. The resulting matrix $\boldsymbol{M}^{c,*}$ satisfies both the unique assignment and equipartition constraints, effectively mapping each point sample $\boldsymbol{x}_i$ in the training batch to its corresponding prototype $\boldsymbol{p}_k^c$ for each class $c$.

**Prototype Update.** In previous solutions [31, 5], the prototypes are typically computed by determining the centers of the corresponding embedded point samples and subsequently updating them based on the online clustering results. However, this strategy may not effectively mitigate the widespread class imbalance issue, resulting in sub-optimal prototypes specifically for rare classes.

To overcome this limitation, we introduce a prototype calibration strategy with a learnable calibration factor, which estimates the representative level of each class prototype showing the importance of tailed class prototype embeddings. The calibration factor is determined by a rectified sigmoid function that adjusts the distance between the prototypes $\boldsymbol{P}^c = \{\boldsymbol{p}_k^c\}_{k=1}^K$ of class $c$ and the point embedding set $\{\boldsymbol{e}_i^c\}_{i=1}^{N_c}$ assigned to $\boldsymbol{p}_k^c$ by prototype association in current training batch:

$$w^c = \frac{1}{N_c K} \sum_{i=1}^{N_c} \sum_{k=1}^{K} \frac{1}{1 + \exp((\boldsymbol{e}^{c_i})^\top \boldsymbol{p}_k^c)}, \quad w^c \in [0, 1]. \tag{11}$$

Thus, the formula for the modified prototype updating method with prototype calibration is as follows:

$$\boldsymbol{p}_k^c \leftarrow \mu \boldsymbol{p}_k^c + w^c (1 - \mu) \bar{\boldsymbol{E}}^c, \tag{12}$$

where $\mu \in [0, 1]$ is a momentum coefficient, and $\bar{\boldsymbol{E}}^c \in \mathbb{R}^D$ is the mean vector of $\{\boldsymbol{e}_i^c\}_{i=1}^{N_c}$.

### 3.3 Supervision Scheme

Based on prototype assignment and update, we get the learned prototypes where $\{\boldsymbol{p}_k^c\}_{k=1}^K$ has class representation ability and is typical for their corresponding class $c$, and $\boldsymbol{p}_k^c$ represents $k^{th}$ property within the class $c$. In order to shape well-structured embedded spaces, we designed a comprehensive supervision scheme based on metric learning to enhance inter-class prototypes separation (see $\mathcal{L}_{\text{PPS}}$ in Eq. 13), intra-class point-prototype compactness (see $\mathcal{L}_{\text{PPC}}$ in Eq. 14), and inter-class point-class separation (see $\mathcal{L}_{\text{PCS}}$ in Eq. 15) through optimizing point-prototype-class distances.

**Prototype-Prototype Distances Optimization.** In order to differentiate between different prototypes within the same class (*i.e.*, *inter-class prototypes separation*), we utilize a prototype-based metric learning strategy to enhance the relationship between points and their assigned prototypes within a specific class. By utilizing the point-to-prototype permutation matrix $\boldsymbol{M}^{c,*}$, this strategy encourages each point embedding $\boldsymbol{e}_i^c$ to be similar to its assigned ('positive') prototype $\boldsymbol{p}_{k_i}^{c_i}$ and dissimilar to other irrelevant ('negative') prototypes $\mathcal{P}^- = \{\boldsymbol{p}_{k_j}^{c_j} | \boldsymbol{p}_{k_j}^{c_j} \in \{\boldsymbol{p}_k^c\}_{c,k=1}^{C,K}$ and $\boldsymbol{p}_{k_j}^{c_j} \neq \boldsymbol{p}_{k_i}^{c_i}\}$:

$$\mathcal{L}_{\text{PPS}} = -\log \frac{\exp((\boldsymbol{e}_i^c)^\top \boldsymbol{p}_{k_i}^{c_i}/\tau)}{\exp((\boldsymbol{e}_i^c)^\top \boldsymbol{p}_{k_i}^{c_i}/\tau) + \sum_{\boldsymbol{p}_{k_j}^{c_j} \in \mathcal{P}^-} \exp((\boldsymbol{e}_i^c)^\top \boldsymbol{p}_{k_j}^{c_j}/\tau)}, \tag{13}$$

where $\tau$ controls the concentration level of the distributions over classes.

**Point-Prototype Distance Optimization.** To reduce intra-cluster variation, we directly minimize the distance between each embedded point $\boldsymbol{e}_i^c$ and its assigned prototype $\boldsymbol{p}_{k_i}^{c_i}$ (*i.e.*, *intra-class point-prototype compactness*), resulting in tighter and more coherent class representations:

$$\mathcal{L}_{\text{PPC}} = (1 - (\boldsymbol{e}_i^c)^\top \boldsymbol{p}_{k_i}^{c_i})^2. \tag{14}$$

**Point-Class Distances Optimization.** In addition to the constraints between point-prototype, for more accurate classification prediction, we aim to correctly associate points with their respective classes (*i.e.*, *inter-class point-class separation*). Specifically, we push the embedded points closer to

their corresponding class (*i.e.*, $c_i$) and enlarge the distance from other irrelevant classes (*i.e.*, $c' \neq c_i$). Given the groundtruth class $c_i$ of each point embedding $\boldsymbol{e}_i$, the cross-entropy loss is used for $\mathcal{L}_{\text{PCS}}$ as:

$$\mathcal{L}_{\text{PCS}} = -\log \frac{\exp(-d_{\boldsymbol{e}_i,c_i})}{\exp(-d_{\boldsymbol{e}_i,c_i}) + \sum_{c' \neq c_i} \exp(-d_{\boldsymbol{e}_i,c'})}, \ \text{ with } \ d_{\boldsymbol{e}_i,c} = \arg\min\{\langle \boldsymbol{e}_i, \boldsymbol{p}_k^c \rangle\}_{k=1}^K \qquad (15)$$

where the point-class distance $d_{\boldsymbol{e}_i,c} \in [-1,1]$ is the distance to the closest prototype of class $c$.
**Total Losses.** Our prototypical classifier can effectively optimize the point embedding space extracted through $\phi$ by minimizing the combined loss over all the training point samples:

$$\mathcal{L}_{\text{Total}} = \mathcal{L}_{\text{PCS}} + \alpha \mathcal{L}_{\text{PPS}} + \beta \mathcal{L}_{\text{PPC}}, \qquad (16)$$

where $\alpha, \beta$ are hyperparameters that control the trade-off between inter-class separation, intra-class compactness, and correct classification. By minimizing $\mathcal{L}_{\text{Total}}$, our model segments the point cloud effectively while producing compact and discriminative class representations in the embedding space.

## 3.4 Implementation Details

We use PointTransformer V2 [2] as the encoder $\phi$ and replace the original MLP-based semantic prediction head with our prototypical classifier. Instance segmentation and panorama segmentation tasks can be viewed as more fine-grained prototype classification problems. For these tasks, we additionally learn fine-grained prototypes $\boldsymbol{P}^{c,o} \in \mathbb{R}^{D \times T}$, $\boldsymbol{P}^c \in \mathbb{R}^{T \times K}$, and a fine-grained permutation matrix $\boldsymbol{M}^{c,o} \in \{0,1\}^{T \times N_c}$, $\boldsymbol{M}^c \in \{0,1\}^{K \times T}$, where $o \in O$, $T, O$ are the number of instance prototypes and instances per class. The association and update of prototypes follow §3.2. Instance-level supervision using a cross-entropy loss is incorporated into our supervision scheme (§3.3).

**Reproducibility**. Training and testing are conducted on eight NVIDIA A100 GPUs. More details and parameter settings can be found in the Appendix.

# 4 Important Knows and Knowledge Gap

**Specialized point cloud segmentation.** Point cloud segmentation comprises three separate tasks, *i.e.*, semantic, instance and panoptic segmentation, each of which focuses on different semantic aspects. Historically, researchers have devised specialized models and optimization objectives for each task.

*Semantic segmentation* is to achieve comprehension of high-level semantic concepts through the grouping of points into discrete semantic units. The advent of point convolution [16] has facilitated the development of sophisticated models, encompassing context aggregation [27, 19], graph convolution integration [24, 17], and contrastive learning [32, 33]. Recently, Transformer architectures [18, 2], which has been successful in vision [34, 35, 36], have obtained considerable research interest.

*Instance segmentation* involves assigning foreground points to individual object instances, similar to video tasks [37], which can be achieved through three main approaches: ① *top-down models* [21, 12] that initially detect object bounding boxes and generate an instance mask for each box, ② *bottom-up models* [38, 39] that learn distinctive point embeddings on instance boundaries, energy levels, geometric structures, and point-center offsets, and group them into instances, and ③ *single-shot models* [22] that directly predict instance masks through a set of learnable object queries.

*Panoptic segmentation* seeks comprehensive scene understanding, considering semantic relations between background points and instance memberships of foreground points. Traditional solutions involve decomposing the problem into manageable tasks such as box-based segmentation [11, 20], thing-stuff merging [40], instance clustering [23, 41, 25]. More recent DETR-like approaches [28] shifted towards an end-to-end scheme using Transformer [42] and achieved compelling performance.

**Prototypical classifier.** Prototype-based classification, an exemplar-driven approach that compares observations with representative examples, has gained particular attention among various machine learning algorithms, which include classical statistics-based methods, Support Vector Machines, and Multilayer Perceptrons. The nearest neighbors rule, the earliest prototype learning method [43], has led to the development of many well-known, non-parametric classifiers [44], such as generalized Learning Vector Quantization (LVQ) [45], and Neighborhood Component Analysis [46]. Metric learning [47, 48] is also naturally related to prototype learning. Recent efforts have attempted to integrate deep learning into prototype learning, demonstrating its potential in few-shot [49], self-supervised learning [50], weakly supervised learning [51, 52], as well as supervised learning [53, 54]

and interpretable networks [55]. Building on these successes, we aim to advance this research by developing a universal 3D segmentation framework based on prototypical classification. Our method, PROTOSEG, incorporates prototype-anchored classification design and simple Transformer architecture to better capture the nature of prototypical learning and effectively handle the heterogeneity across various segmentation tasks using the same architecture.

**Universal 3D Segmentation.** Universal segmentation strives to adopt a unified architecture capable of addressing diverse segmentation tasks. One of the early attempts at unification through multi-task learning is MT-PNet [56], which uses two branches to predict semantic class labels and instance embedding labels, respectively. More recent studies, such as those presented in [57], employ clustering schemes to integrate semantic and instance information, but their heuristic post-processing technique (*e.g.*, mean shift) can bring serious computational burden. In comparison to these pioneering efforts, PROTOSEG distinguishes itself by being: ① more transparent - leveraging straightforward, case-based reasoning for 3D tasks; ② more flexible - handling multiple segmentation tasks simultaneously; and ③ more powerful - leading specialized methods by notable margins.

**Multi-Task Image Segmentation** Multi-Task Image Segmentation aims to develop a cohesive architecture to address various segmentation challenges. K-Net [58] pioneered this approach, leveraging dynamic kernel learning for mask generation. In recent developments, several architectures inspired by DETR [59] have formulated different tasks within a mask classification paradigm. For instance, Max-DeepLab [60] and kMaX-DeepLab [61] negate the need for box predictions by employing conditional convolutions, bridging the gap between box-dependent and box-independent methods for the first time. MaskFormer [62] and its subsequent versions [63, 64] introduce a streamlined and effective inference strategy to merge outputs into a task-specific prediction format using a collection of binary masks. In addition, methods based on dense prediction [65], optimal transport [66], graph modeling [67], and clustering [68, 69, 70] also have achieved great success.

Unlike the above methods, our approach perceives these three closely-related tasks as a prototype-anchored classification challenge with distinct granularity levels. We define prototypes as class sub-centroids derived from the feature embeddings of training samples. Then, a test sample is directly classified based on its proximity to the nearest centroids. Such an approach, rooted in case-based reasoning, introduces a distinct element of ad-hoc interpretability to our method (see Fig. 3 in §5.4).

# 5 Empirical Evidence

PROTOSEG is the first framework to support three core point cloud segmentation tasks with a single unified architecture. To demonstrate its broad applicability and wide benefit, we conduct extensive experiments on semantic (§5.1), instance (§5.2), panoptic (§5.3) segmentation, and carry out ablation studies (§5.4) related to our framework design. More qualitative results are available in Appendix.

## 5.1 Experiment on Semantic Segmentation

**Dataset.** **S3DIS** [9], a large-scale indoor point cloud dataset, encompasses point clouds from 271 rooms across 6 areas. Each room represents a medium-sized point cloud, annotating every point with a semantic label from one of the 13 classes. **ScanNet V2** [10] provides over 1,500 indoor scenes and around 2.5 million annotated RGB-D images with approximately 90% surface coverage. The benchmark is evaluated on 20 semantic classes, which include 18 different object classes. **SemanticKITTI** [11] is introduced based on the well-known KITTI Vision [71] benchmark illustrating complex outdoor traffic scenarios. It contains 22 data sequences, 43,552 frames of outdoor scenes, of which 23,201 frames with panoptic labels are used for training and validation, and the remaining 20,351 frames without labels are used for testing. There are annotated point-wise labels in 20 classes for segmentation tasks, 8 of which are defined as thing classes.

**Metric.** To evaluate our semantic segmentation performance, we apply the mean Intersection-over-Union (mIoU ↑), the overall accuracy (OAcc ↑) taking all points into consideration and the average class accuracy (mAcc ↑) of all semantic classes upon the whole dataset.

**Performance Comparison.** Table 1 and Table 2 show the results of our PROTOSEG model compared with previous methods on S3DIS [9] and ScanNet v2 [10], respectively. Our model outperforms prior methods across all evaluation metrics. Notably, PROTOSEG significantly outperforms PTv2 [2] by **0.9%** and **1.2%** mIoU on the ScanNet v2 [10] validation and test set, and

Table 1: Comparisons of **semantic segmentation** performance on S3DIS [9] Area 5 (see §5.1).

| Method | OAcc | mAcc | mIoU |
|---|---|---|---|
| JSNet [57] [AAAI'20] | 87.7 | 61.4 | 54.5 |
| SegGCN [72] [CVPR'20] | 88.2 | 70.4 | 63.6 |
| SCF-Net [73] [CVPR'21] | 88.4 | 71.6 | 82.7 |
| PAConv [74] [CVPR'21] | – | - | 66.6 |
| RepSurf-U [75] [CVPR'22] | 90.2 | 76.0 | 68.9 |
| CBL [33] [CVPR'22] | 90.6 | 75.2 | 69.4 |
| PTv1 [18] [ICCV'21] | 90.8 | 76.5 | 70.4 |
| FastPT [76] [CVPR'22] | – | 77.9 | 70.3 |
| PointMixer [77] [ECCV'22] | – | 77.4 | 71.4 |
| PTv2 [2] [NeurIPS'22] | 91.1 | 77.9 | 71.6 |
| StratifiedFormer [78] [CVPR'22] | 91.5 | 78.1 | 72.0 |
| **Ours (Area 5)** | **92.2** | **78.6** | **72.3** |

Table 2: Comparisons of **semantic segmentation** with mIoU on ScanNet v2 [10] (see §5.1).

| Method | Test | Val. |
|---|---|---|
| SegGCN [72] [CVPR'20] | 58.9 | – |
| RandLA-Net [79] [CVPR'20] | 64.5 | – |
| PointASNL [27] [CVPR'20] | 66.6 | 63.5 |
| RPNet [80] [ICCV'21] | 68.2 | – |
| FusionNet [81] [ECCV'20] | 68.8 | – |
| JSENet [82] [ECCV'20] | 69.9 | – |
| RepSurf-U [75] [CVPR'22] | 70.2 | – |
| PTv1 [18] [ICCV'21] | - | 70.6 |
| PointNeXt [19] [NeurIPS'22] | 71.2 | 71.5 |
| StratifiedFormer [78] [CVPR'22] | 73.7 | 74.3 |
| PTv2 [2] [NeurIPS'22] | 75.2 | 75.4 |
| **Ours** | **76.4** | **76.3** |

surpasses other methods in all metrics on the S3DIS [9] dataset. These results further support the robustness and effectiveness of our method for large-scale point cloud semantic segmentation tasks.

Notably, our method yields a more substantial performance enhancement on ScanNet V2 than on S3DIS compared to the main competitor PTv2 [2]. This discrepancy can be attributed to PROTOSEG's statistical characteristics, which can mine richer intra-class variations on the larger scale and more categories ScanNet V2. To evaluate the performance more comprehensively, we report the results of semantic segmentation on the SemanticKITTI validation set. PROTOSEG achieves a mIou of 74.2%, confirming the performance benefits of our model.

Table 3: Comparisons of **semantic segmentation** performance on SemanticKITTI *val* set (see §5.1).

| Method | mIoU |
|---|---|
| Cylinder3D [83][CVPR'21] | 65.9 |
| 2DPASS [84][ECCV'22] | 69.3 |
| $(AF)^2$-S3Net [85][CVPR'21] | 74.2 |
| Ours | 74.2 |

## 5.2 Experiment on Instance Segmentation

**Dataset.** S3DIS [9] and ScanNet V2 [10] are utilized for instance segmentation experiments.

**Metric.** We use mean coverage (mCov ↑), mean weighed coverage (mWCov ↑), mean precision (mPrec ↑), and mean recall (mRec ↑) as the evaluation metrics for S3DIS [9]. Additionally, ScanNet V2 [10] is evaluated using mean average precisions (mAPs ↑) under different IoU thresholds.

**Performance Comparison.** As is depicted in Tables 4-5, our method significantly outperforms other methods on the S3DIS [9] dataset. The main criteria, mCov and mWCov, take into account both the completeness and accuracy of instance segmentation results. Specifically, in terms of the main criteria, our method yields a mCov of **66.8%**, a mWCov of **68.4%**, a mPrec of **69.7%** and a mRec of **66.3%** in Area 5, surpassing most methods. Furthermore, the superior performance on ScanNet V2 [10] dataset in both the *val.* and *test* sets also highlights the effectiveness of our approach for instance segmentation. The significant improvements across various metrics and benchmarks demonstrate the efficacy and robustness of PROTO-SEG, even in challenging scenarios.

Table 4: Comparisons of **instance segmentation** performance on S3DIS [9] Area 5 (see §5.2 for more details).

| Method | mCov | mWCov | mPrec | mRec |
|---|---|---|---|---|
| PointGroup [86] [CVPR'20] | – | – | 61.9 | 62.1 |
| MaskGroup [39] [ICME'22] | – | – | 62.9 | 64.7 |
| SSTNet [22] [ICCV'21] | 42.7 | 59.3 | 65.5 | 64.2 |
| DyCo3D [87] [CVPR'21] | 63.5 | 64.6 | 64.3 | 64.2 |
| HAIS [38] [ICCV'21] | 64.3 | 66.0 | **71.1** | 65.0 |
| DKNet [88] [ECCV'22] | 64.7 | 65.6 | 70.8 | 65.3 |
| **Ours** | **66.8** | **68.4** | 69.7 | **66.3** |

Table 5: Comparisons of **instance segmentation** performance on ScanNet v2 [10] (see §5.2 for more details).

| Module | Test | | Val. | |
|---|---|---|---|---|
| | mAP | mAP$_{50}$ | mAP | mAP$_{50}$ |
| DyCo3D [87] [CVPR'21] | 39.5 | 64.1 | 35.4 | 57.6 |
| PointGroup [86] [CVPR'20] | 40.7 | 63.6 | 34.8 | 56.7 |
| MaskGroup [39] [ICME'22] | 43.4 | 66.4 | 42.0 | 63.3 |
| HAIS [38] [ICCV'21] | 45.7 | 69.9 | 43.5 | 64.1 |
| SoftGroup [12] [CVPR'22] | 50.4 | 76.1 | 46.0 | 67.6 |
| SSTNet [22] [ICCV'21] | 50.6 | 69.8 | **49.4** | 64.3 |
| **Ours** | **51.2** | **78.1** | 47.8 | **68.4** |

## 5.3 Experiment on Panoptic Segmentation

**Dataset.** SemanticKITTI [11] is utilized for panoptic segmentation experiments.

Table 6: Comparisons of **panoptic segmentation** performance on SemanticKITTI [11] test (see §5.3).

| Method | PQ | mIou | PQ$^\dagger$ | RQ | SQ | PQ$^{Th}$ | RQ$^{Th}$ | SQ$^{Th}$ | PQ$^{St}$ | RQ$^{St}$ | SQ$^{St}$ |
|---|---|---|---|---|---|---|---|---|---|---|---|
| LPSAD [90] [IROS'20] | 38.0 | 50.9 | 47.0 | 48.2 | 76.5 | 25.6 | 31.8 | 76.8 | 47.1 | 60.1 | 76.2 |
| Panoster [91] [RAL'21] | 52.7 | 59.9 | 59.9 | 64.1 | 80.7 | 49.4 | 58.5 | 83.3 | 55.1 | 68.2 | 78.8 |
| Panoptic-PolarNet [23] [CVPR'21] | 54.1 | 59.5 | 60.7 | 65.0 | 81.4 | 53.3 | 60.6 | 87.2 | 54.8 | 68.1 | 77.2 |
| DS-Net [41] [CVPR'21] | 55.9 | 61.6 | 62.5 | 66.7 | 82.3 | 55.1 | 62.8 | 87.2 | 56.5 | 69.5 | 78.7 |
| EfficientLPS [40] [TR'21] | 57.4 | 61.4 | 63.2 | 68.7 | 83.0 | 53.1 | 60.5 | 87.8 | 60.5 | 74.6 | 79.5 |
| GP-S3Net [25] [ICCV'21] | 60.0 | **70.8** | 69.0 | 72.1 | 82.0 | 65.0 | 74.5 | 86.6 | 56.4 | 70.4 | 78.7 |
| SCAN [28] [AAAI'22] | 61.5 | 67.7 | 67.5 | 72.1 | 84.5 | 61.4 | 69.3 | 88.1 | 61.5 | 74.1 | 81.8 |
| Panoptic-PHNet [15] [CVPR'22] | 61.5 | 66.0 | 67.9 | 72.1 | **84.8** | 63.8 | 70.4 | **90.7** | 59.9 | 73.3 | **80.5** |
| **Ours** | **62.4** | 67.5 | **68.5** | **74.4** | 83.4 | **65.6** | **72.6** | 89.1 | **61.8** | **75.7** | 79.2 |

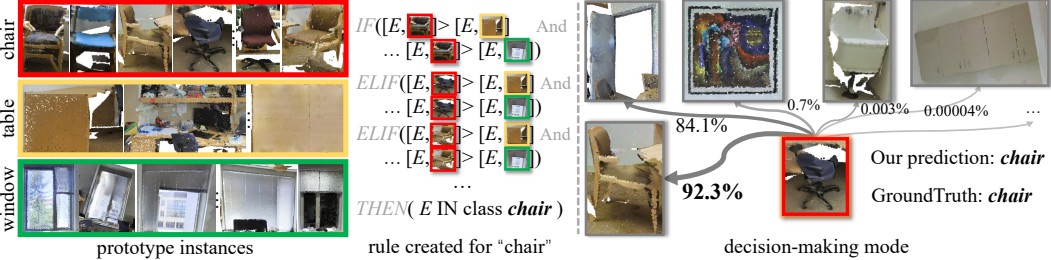

Figure 3: Interpretable prototype (*i.e.*, class sub-centroids) with *IF*···*Then* rules (left) and decision-making process based on distance to prototypes (right). ▢▢▢ colors represent prototype instance for three S3DIS [9] classes, [·,·] stands for distance between point embedding $E$ and instances.

**Metric.** We use the panoptic quality (PQ) [89] as our main metric to evaluate the performance of panoptic segmentation. PQ can be seen as the multiplication of segmentation quality (SQ) and recognition quality (RQ). These three metrics can be extended to things and stuff classes, denoted as $PQ^{Th}$, $PQ^{St}$, $RQ^{Th}$, $RQ^{St}$, $SQ^{Th}$, $SQ^{St}$, respectively. $PQ^\dagger$ replaces PQ with IoU for stuff classes.

**Performance Comparison.** Table 6 presents a comparison of panoptic segmentation performance on the SemanticKITTI [11] test. Our proposed method demonstrates a significant improvement over the best-performing baseline method, with a **62.4%** increase in PQ. It is noteworthy that our unified framework demonstrates superior performance in both things and stuff classes, as evidenced by outperforming the second-best method, Panoptic-PHNet [15], in 7 out of 11 metrics.

Overall, PROTOSEG demonstrates excellent performance across various segmentation tasks (see §5.1-5.3) on both indoor and outdoor datasets. Notably, it does not require inherent modifications to the architecture or the training regime, which highlights its generality and flexibility.

## 5.4 Diagnostic Experiment

This section conducts ablation studies for PROTOSEG's semantic segmentation on S3DIS [9] Area5.

**Study of Ad-hoc Explainability.** In §2 and §5.1-5.3, we have explained the simple yet transparent mechanism of PROTOSEG and verified its robustness and effectiveness in representation learning. We now highlight PROTOSEG's interpretability in Fig. 3, which depicts prototypes that represent specific objects in the training set. Fig. 3 (left) displays representative instances, serving as interpretable prototypes (class sub-centroids) for three S3DIS [9] classes. These objects, diverse in appearance, viewpoints, illuminations, and scales, characterize their respective classes, providing a human-interpretable overview. Leveraging the simplicity of the Nearest Centroids mechanism, we formulate intuitive *IF*···*Then* rules, providing an intuitive glimpse into PROTOSEG's decision-making process. Furthermore, we can elucidate PROTOSEG's predictions by visualizing the computed distance (similarity) between the query and prototypes. As illustrated in Fig. 3 (right), the model accurately classifies an observation, perceiving it as more closely resembling a particular ***chair*** exemplar (see the top right of Fig. 2). Overall, this visualization clearly shows that the classification decision procedure relies on the notion of distance beyond the selection of the encoder $\phi$.

**Key Components.** We first investigate our Prototypical Classifier (§3.2) and supervision scheme (§3.3) on S3DIS [9] and ScanNet V2 [10] datasets. As shown in Table 7a, introducing prototypical classifiers in conjunction with $\mathcal{L}_{PCS}$ improves the performance to 71.74% and 75.84%. Adding either $\mathcal{L}_{PPS}$ or $\mathcal{L}_{PPC}$ to the prototypical classifiers and $\mathcal{L}_{PCS}$ promotes performance gains, indicating the

Table 7: A set of **ablative studies** (see §5.4) on S3DIS [9] Area 5.

(a) Key Components

| Prototypical Classifiers | $\mathcal{L}_{PCS}$ (Eq. 15) | $\mathcal{L}_{PPS}$ (Eq. 13) | $\mathcal{L}_{PPC}$ (Eq. 14) | S3DIS (mIoU) | ScanNetv2 (mIoU) |
|---|---|---|---|---|---|
| | ✓ | | | 71.53 | 75.36 |
| ✓ | ✓ | | | 71.74 | 75.84 |
| ✓ | ✓ | ✓ | | 72.05 | 76.07 |
| ✓ | ✓ | | ✓ | 72.06 | 76.12 |
| ✓ | ✓ | ✓ | ✓ | 72.34 | 76.32 |

(b) Prototype Number $K$

| # Prototype | mIoU (%) |
|---|---|
| $K = 1$ | 71.48 |
| $K = 5$ | 71.84 |
| $K = 10$ | 72.34 |
| $K = 20$ | 72.18 |
| $K = 50$ | 72.05 |

(c) Distance Measure

| Distance | mIoU (%) |
|---|---|
| Standard | 72.10 |
| Huberized | 72.17 |
| Cosine | 72.34 |

(d) Momentum Coefficient

| Coefficient $\mu$ | mIoU (%) |
|---|---|
| 0.9 | 71.62 |
| 0.99 | 71.88 |
| 0.999 | 72.34 |
| 0.9999 | 72.04 |

(e) Class Balance

| Strategy | mIoU (%) |
|---|---|
| w/o | 71.49 |
| $w^c$ | 72.34 |
| $M_p$ | 72.14 |
| $M_n$ | 71.83 |

(f) Sinkhorn Iterations $S$

| Iterations (Eq. 10) | mIoU (%) |
|---|---|
| $S = 1$ | 71.29 |
| $S = 3$ | 72.34 |
| $S = 5$ | 71.43 |

value of explicitly learning point-prototype relations. The improvements observed by introducing prototypical classifiers and supervision scheme (mIoU: $71.53\% \rightarrow 72.34\%$ on S3DIS [9], $75.36\% \rightarrow 76.32\%$ on ScanNet V2 [10]) confirm the effectiveness of our proposed framework.

**Prototype Number Per Class.** We then analyze the impact of the number of prototypes per class on segmentation performance, as reported in Table 7b. With a single prototype ($K=1$), each class is represented by the mean embedding of its point samples, yielding the mIoU score of $71.48\%$. The mIoU score peaks at $72.34\%$ when $K = 10$. Further increment of prototypes to $K = 20$, $K = 50$ only results in a slight decrease in performance, with mIoU scores of $72.34\% \rightarrow 72.18\% \rightarrow 72.05\%$, respectively. These results suggest an optimal balance between accuracy and computational cost at $K = 10$ prototypes per class, affirming our rationale for leveraging multiple prototypes to encapsulate intra-class variations and enhance segmentation performance. It would be interesting to find ways to ascertain this number automatically. In our exploration, we encountered clustering techniques like [92] that might help, but complex algorithms led to time challenges.

**Distance Measure.** Next, we examine the impact of different distance metrics on our Supervision Scheme (see Eqs. 13-15 in §3.3). By default, we use cosine distance (denoted as *Cosine* in Table 7c). We also consider two alternative metrics, *i.e.*. standard Euclidean distance (*Standard*) and Huber-like function [93] (*Huberized*). Table 7c demonstrates that the *Cosine* distance surpasses the unnormalized Euclidean metrics, affirming its superiority in measuring point-prototype similarity.

**Momentum Coefficient.** Table 7d quantifies the effect of the momentum coefficient ($\mu$ in Eq. 12). Our model achieves the best performance when the momentum coefficient is set to 0.999. As the momentum coefficient decreases, the performance gradually declines. These results suggest that a higher value (*i.e.*, slower updating) generally leads to better results in the common case.

**Class Balance Strategy.** Next, we proceed to validate the effectiveness of our strategy in mitigating the issue of category imbalance. As Table 7e shown, the baseline (*w/o*) fixes $w^c = 1$ in Eq. 12, $M_l$ [31] and $M_u$ [94] utilize parameter and non-parametric augmented memory banks, respectively. The reported results provide evidence for the efficacy of our prototype calibration strategy.

**Sinkhorn-Knopp Iteration.** Finally, we verify the effect of iterations on the model. Table 7f shows that the optimal segmentation results can be achieved using a smaller number ($S = 3$) of iterations. This further verifies the efficiency of our Prototype Association (see §3.2).

## 6 Conclusion

We present PROTOSEG, a novel, prototype-anchored classification framework for 3D point cloud segmentation that unifies semantic, instance, and panoptic segmentation tasks. By considering these tasks under the umbrella of classification problems but at various granularity, PROTOSEG offers an elegant, general, and flexible solution that eschews task-specific network architectures. Compared to task-specific models, PROTOSEG offers several advantages: i) The prototype-based classification mechanism is simple and transparent. ii) Our supervision scheme, which leverages prototype-based metric learning, exhibits robust representational learning ability. iii) PROTOSEG offers ad-hoc explainability by anchoring class exemplars to real observations. Our comprehensive experiments demonstrate the superior performance of PROTOSEG in terms of efficacy and enhanced interpretability. Overall, we believe that our work may motivate further research in this field.

**Acknowledgement.** This work was supported in part by the National Natural Science Foundation of China (No. 62176139, 62106128, 62176141), the Major basic research project of Shandong Natural Science Foundation (No. ZR2021ZD15), the Natural Science Foundation of Shandong Province (No. ZR2021QF001), the Young Elite Scientists Sponsorship Program by CAST (No. 2021QNRC001), the Open project of Key Laboratory of Artificial Intelligence, Ministry of Education, the Shandong Provincial Natural Science Foundation for Distinguished Young Scholars (ZR2021JQ26), the Taishan Scholar Project of Shandong Province (tsqn202103088).

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
