# Unified 3D Segmenter As Prototypical Classifiers
## *Appendix*

**Zheyun Qin**[1*], **Cheng Han**[2*], **Qifan Wang**[3], **Nie Xiushan**[4], **Yilong Yin**[1†], **Xiankai Lu**[1†]

[1]Shandong University, [2]Rochester Institute of Technology, [3]Meta AI, [4]Shandong Jianzhu University

This appendix contains additional details for the NeurIPS 2023 submission, organized as follows:

- §1 provides the pseudo code of PROTOSEG.

- §2 offers more details about instance segmentation and panorama segmentation.

- §3 introduces experimental settings and more quantitative results on segmentation.

- §4 depicts more studies regarding the ad-hoc explainability of PROTOSEG.

- §5 explores more ablation experiments.

- §6 plots qualitative semantic, instance segmentation results.

- §7 discusses our limitations, social impact and points out several directions of future work.

## 1 Pseudo Code of PROTOSEG and Code Release

The pseudo-code of PROTOSEG is given in Algorithm 1. To guarantee reproducibility, our full implementation shall be publicly released upon paper acceptance.

## 2 Application of PROTOSEG in Instance and Panorama Segmentation

For these tasks, we additionally learn fine-grained prototypes $\boldsymbol{P}^{c,o} \in \mathbb{R}^{D \times T}$, $\boldsymbol{P}^c \in \mathbb{R}^{T \times K}$, where $\boldsymbol{p}_t^{c,o} \in \mathbb{R}^D$ denotes the center of the $t^{th}$ sub-cluster (property) of training point samples belonging to instance $o$ of class $c$. Similarity to Eq.5, we make predictions by comparing point embedding $\boldsymbol{e}_i \in \mathbb{R}^D$, with prototype $\boldsymbol{p}_t^{c,o}$ and assigning the corresponding prototype's class as the response:

$$p(c, o | \boldsymbol{x}_i) = \frac{\exp(-d_{\boldsymbol{e}_i, c, o})}{\sum_{c'=1, o'=1}^{C, O} \exp(-d_{\boldsymbol{e}_i, c', o'})}, \text{ with } d_{\boldsymbol{e}_i, c, o} = \arg\min\{\langle \boldsymbol{e}_i, \boldsymbol{p}_t^{c,o} \rangle\}_{t=1}^T, \tag{1}$$

where the negative cosine distance measure $\langle \cdot, \cdot \rangle$ is defined as $\langle \boldsymbol{e}_i, \boldsymbol{p}_t^{c,o} \rangle = -\boldsymbol{e}_i^\top \boldsymbol{p}_t^{c,o}$.

To get the informative prototype that can represent the properties of the instance and class, we learn a fine-grained permutation matrix $M$ for prototype assignment and update. Similarity to Eq.6, we aim to maximize the similarities $J(M)$:

$$\max_{\boldsymbol{M}} \quad \mathcal{J}(\boldsymbol{M}) = \text{Tr}((\boldsymbol{M}^{c,o})^\top (\boldsymbol{P}^{c,o})^\top \boldsymbol{E}^c), \tag{2}$$

where $\text{Tr}(\cdot)$ denotes the matrix trace, $\boldsymbol{M}^{c,o} = \{\boldsymbol{m}_{t,i}^{c,o}\}_{t,i=1}^{T,N_c} \in \{0,1\}^{T \times N_c}$ is a point-to-prototype permutation matrix that denotes the association between point and prototype, $\boldsymbol{m}_{t,i}^{c,o} \in \{0,1\}$ denotes the one-hot assignment vector that assigns the point sample $\boldsymbol{x}_i$ to the prototype $t$ of instance $o$ of class $c$. The association and update of prototypes follow Section 3.2. Instance-level supervision using a cross-entropy loss is incorporated into our supervision scheme (Section 3.3).

---

[*]Equal contribution. [†]Corresponding authors.

37th Conference on Neural Information Processing Systems (NeurIPS 2023).

# 3 More Experiments on Point Cloud Segmentation

In this section, we show experimental settings on different segmentation tasks and datasets in table. 1, as well as comparisons with more methods for semantic and instance segmentation tasks in tables 2-5.

Table 1: Experimental Settings on different segmentation tasks and datasets. We denote Lr=initial learning rate.

| Task | Dataset | Lr | Optimizer | Decay | # $K$ | # $O$ | # $T$ | # $\mu$ | # $\tau$ | # $\kappa$ | # $\alpha$ | # $\beta$ |
|------|---------|-----|-----------|-------|------|------|------|-------|-------|--------|--------|--------|
| Semantic *Seg.* | S3DIS [1] | $1 \times 10^{-3}$ | AdamW | 0.5 | 10 | - | - | 0.999 | 0.1 | 0.05 | 0.01 | 0.01 |
| Semantic *Seg.* | ScanNet V2 [2] | $1 \times 10^{-3}$ | AdamW | 0.02 | 8 | - | - | 0.999 | 0.1 | 0.05 | 0.02 | 0.01 |
| Semantic *Seg.* | SemanticKITTI [3] | $1 \times 10^{-3}$ | AdamW | 0.2 | 8 | - | - | 0.999 | 0.1 | 0.05 | 0.01 | 0.01 |
| Instance *Seg.* | S3DIS [1] | $1 \times 10^{-3}$ | AdamW | 0.02 | 10 | 15 | 15 | 0.99 | 0.1 | 0.03 | 0.05 | 0.02 |
| Instance *Seg.* | ScanNet V2 [2] | $1 \times 10^{-2}$ | AdamW | 0.005 | 8 | 60 | 12 | 0.999 | 0.1 | 0.05 | 0.1 | 0.05 |
| Panoptic *Seg.* | SemanticKITTI [3] | $1 \times 10^{-3}$ | AdamW | 0.01 | 8 | 50 | 12 | 0.999 | 0.1 | 0.05 | 0.1 | 0.01 |

# 4 Additional Study of Ad-hoc Explainability

In our principal study on Ad-hoc explainability (see §5.4 in the main paper), we maintained the use of PROTOSEG with Transformer [27] as the encoder, adhering to the same parameter settings as in the optimal model, across a training span of 180 epochs. During the initial 150 epochs, we employed standard training procedures, wherein class sub-centroids were computed as prototypes. Subsequent to this phase, each sub-centroid was anchored to its closest training points, guided by the cosine similarity of the embeddings. In the concluding 30 epochs, the prototypes underwent updates exclusively in accordance with the embeddings of their corresponding anchored training points. As a result, the prototype instance depicted in Fig.3 of the main paper represents the partial point cloud that is most similar to the class sub-centroids. This approach allowed us to derive a more interpretative version of PROTOSEG, denoted as PROTOSEG$^\dagger$.

**Performance with Improved Interpretability.** We then present the performance score of our PROTOSEG, evaluated based on the interpretable class representatives on S3DIS [1] Area 5. As illustrated in Table 6, the strategy of enforcing the prototypes to be real training points results in only a minor performance decline (OAcc: $92.2\% \rightarrow 91.2\%$). However, this slight reduction is compensated by the advantage of enhanced model interpretability. Even more notably, our more explainable variant, PROTOSEG$^\dagger$, surpasses the performance of the standard Transformer+MLP model (OAcc: $91.2\% \rightarrow 90.8\%$). This result attests to the effectiveness of our method in improving both the performance and interpretability of semantic segmentation models.

**Explain Inner Decision-Making Mode based on *IF* $\cdots$ *Then* Rules.** With the simple Nearest Centroids mechanism, we can use the representative points to form a set of *IF* $\cdots$ *Then* rules [56], so as to intuitively interpret the inner decision-making mode of PROTOSEG for human users. In particular, let $\hat{E}$ denote a sub-centroid point for class $c$, $\check{E}_{1:T}$ representative points for all the other classes, and $E$ is a query point. One linguistic logical *IF* $\cdots$ *Then* rule can be generated for $\hat{I}$:

Table 6: Comparisons of semantic segmentation on S3DIS [1] Area 5.

| Architecture | OACC (%) ↑ |
|-------------|-----------|
| PROTOSEG | 92.2 |
| PROTOSEG$^\dagger$ | 91.2 |
| Transformer+MLP | 90.8 |

$$IF \left([E, \hat{E}] > [E, \check{E}_1] \; AND \; [E, \hat{E}] > [E, \check{E}_2] \; AND \cdots AND \; [E, \hat{E}] > [E, \check{E}_T]\right) \; THEN \; (\text{class } c), \quad (3)$$

where $[\cdot, \cdot]$ stands for similarity, given by PROTOSEG. The final rule for class $c$ is created by combining all the rules of $K$ sub-centroid points $\hat{E}_{1:K}$ of class $c$:

$$
\begin{aligned}
& IF \left([E, \hat{E}_1] > [E, \check{E}_1] \; AND \cdots AND \; [E, \hat{E}_1] > [E, \check{E}_T]\right) \\
& \quad ELIF \left([E, \hat{E}_2] > [E, \check{E}_2] \; AND \cdots AND \; [E, \hat{E}_2] > [E, \check{E}_T]\right) \quad (4) \\
& \quad ELIF \cdots ELIF \left([E, \hat{E}_K] > [E, \check{E}_K] \; AND \cdots AND \; [E, \hat{E}_K] > [E, \check{E}_T]\right) \; THEN \; (\text{class } c).
\end{aligned}
$$

Please note that the interpretability of a classifier mainly comes from its decision-making mode, *i.e.*, a test sample is directly classified into the class with the closest centroids instead of the training strategy or datasets. Therefore, *IF* $\cdots$ *Then* applies to both indoor (S3DIS [1], ScanNet V2 [2]) and outdoor datasets (SemanticKITTI [3]).

Table 2: Comparisons of **semantic segmentation** performance on S3DIS [1] Area 5 (see §5.1).

| Method | OAcc | mAcc | mIoU |
|---|---|---|---|
| PointNet [4] [CVPR'17] | – | 49.0 | 41.1 |
| SegCloud [5] [3DV'17] | – | 57.4 | 48.9 |
| TanConv [6] [CVPR'18] | – | 62.2 | 52.6 |
| PointCNN [7] [NeurIPS'18] | 85.9 | 63.9 | 57.3 |
| PointWeb [8] [CVPR'19] | 87.0 | 66.6 | 60.3 |
| HPEIN [9] [CVPR'19] | 87.2 | 68.3 | 61.9 |
| GACNet [10] [CVPR'19] | 87.8 | - | 62.9 |
| PAT [11] [CVPR'19] | – | 70.8 | 60.1 |
| ParamConv [12] [CVPR'18] | – | 67.0 | 58.3 |
| SPGraph [13] [CVPR'18] | 86.4 | 66.5 | 58.0 |
| ASIS [14] [CVPR'19] | 86.9 | 60.9 | 53.4 |
| JSNet [15] [AAAI'20] | 87.7 | 61.4 | 54.5 |
| GACNet [10] [CVPR'19] | 87.8 | – | 62.8 |
| SSP+SPG [16] [CVPR'19] | 87.9 | 68.2 | 61.7 |
| SegGCN [17] [CVPR'20] | 88.2 | 70.4 | 63.6 |
| SCF-Net [18] [CVPR'21] | 88.4 | 71.6 | 82.7 |
| MinkUNet [19] [CVPR'19] | – | 71.7 | 65.4 |
| PAConv [20] [CVPR'21] | – | - | 66.6 |
| KPConv [21] [ICCV'19] | – | 72.8 | 67.1 |
| PointWeb [8] [CVPR'19] | 87.0 | 66.6 | 60.3 |
| RepSurf-U [22] [CVPR'22] | 90.2 | 76.0 | 68.9 |
| CBL [23] [CVPR'22] | 90.6 | 75.2 | 69.4 |
| PTv1 [24] [ICCV'21] | 90.8 | 76.5 | 70.4 |
| FastPT. [25] [CVPR'22] | – | 77.9 | 70.3 |
| PointMixer [26] [ECCV'22] | – | 77.4 | 71.4 |
| PTv2 [27] [NeurIPS'22] | 91.1 | 77.9 | 71.6 |
| StratifiedFormer [28] [CVPR'22] | 91.5 | 78.1 | 72.0 |
| SAT [29] [arXiv'23] | – | 78.8 | 72.6 |
| **Ours (Area 5)** | **92.2** | **78.6** | **72.3** |

Table 3: Comparisons of **semantic segmentation** with mIoU on ScanNet v2 [2] (see §5.1).

| Method | Test | Val. |
|---|---|---|
| PointNet++ [30] [NeurIPS'17] | 55.7 | 53.5 |
| PointEdge [9] [ICCV'19] | 61.8 | 63.4 |
| 3DMV [31] [ECCV'18] | 48.4 | – |
| PanopticFusion [32] [IROS'19] | 52.9 | – |
| PointCNN [7] [NeurIPS'18] | 45.8 | – |
| PointConv [33] [CVPR'19] | 66.6 | 61.0 |
| PointASNL [34] [CVPR'20] | 66.6 | 63.5 |
| JointPointBased [35] [3DV'19] | 63.4 | 69.2 |
| SegGCN [17] [CVPR'20] | 58.9 | – |
| RandLA-Net [36] [CVPR'20] | 64.5 | – |
| KPConv [21] [ICCV'19] | 68.6 | 69.2 |
| RPNet [37] [ICCV'21] | 68.2 | – |
| JSENet [38] [ECCV'20] | 69.9 | – |
| FusionNet [39] [ECCV'20] | 68.8 | – |
| RepSurf-U [22] [CVPR'22] | 70.2 | – |
| SparseConvNet [40] [CVPR'18] | 72.5 | 69.3 |
| PTv1 [24] [ICCV'21] | - | 70.6 |
| PointNeXt [41] [NeurIPS'22] | 71.2 | 71.5 |
| MinkowskiNet [19] [CVPR'19] | 73.6 | 72.2 |
| MinkUNet [19] [CVPR'19] | 73.6 | 72.2 |
| StratifiedFormer [28] [CVPR'22] | 73.7 | 74.3 |
| PTv2 [27] [NeurIPS'22] | 75.2 | 75.4 |
| **Ours** | **76.4** | **76.3** |

Table 4: Comparisons of **instance segmentation** performance on S3DIS [1] Area 5 (see §5.2 for more details).

| Method | mCov | mWCov | mPrec | mRec |
|---|---|---|---|---|
| SGPN [42] [CVPR'18] | 32.7 | 35.5 | 36.0 | 28.7 |
| ASIS [14] [CVPR'19] | 44.6 | 47.8 | 55.3 | 42.4 |
| JSNet [15] [AAAI'20] | 48.7 | 51.5 | 62.1 | 46.9 |
| 3D-Bonet [43] [NeurIPS'19] | – | – | 57.5 | 40.2 |
| PointGroup [44] [CVPR'20] | – | – | 61.9 | 62.1 |
| MaskGroup [45] [ICME'22] | – | – | 62.9 | 64.7 |
| SSTNet [46] [ICCV'21] | 42.7 | 59.3 | 65.5 | 64.2 |
| DyCo3D [47] [CVPR'21] | 63.5 | 64.6 | 64.3 | 64.2 |
| HAIS [48] [ICCV'21] | 64.3 | 66.0 | **71.1** | 65.0 |
| DKNet [49] [ECCV'22] | 64.7 | 65.6 | 70.8 | 65.3 |
| **Ours** | **66.8** | **68.4** | 69.7 | **66.3** |

Table 5: Comparisons of **instance segmentation** performance on ScanNet v2 [2] (see §5.2 for more details).

| Module | Test | | Val. | |
|---|---|---|---|---|
| | mAP | mAP$_{50}$ | mAP | mAP$_{50}$ |
| SGPN [42] [CVPR'18] | – | – | 4.9 | 14.3 |
| GSPN [50] [CVPR'19] | 19.3 | 37.8 | – | 30.6 |
| 3D-SIS [51] [CVPR'19] | – | 18.7 | 16.1 | 38.2 |
| 3D-Bonet [43] [NeurIPS'19] | – | – | 25.3 | 48.8 |
| MTML [52] [ICCV'19] | 20.3 | 40.2 | 28.2 | 54.9 |
| 3D-MPA [53] [CVPR'20] | 35.5 | 61.1 | 35.5 | 59.1 |
| DyCo3D [47] [CVPR'21] | 39.5 | 64.1 | 35.4 | 57.6 |
| PointGroup [44] [CVPR'20] | 40.7 | 63.6 | 34.8 | 56.7 |
| MaskGroup [45] [ICME'22] | 43.4 | 66.4 | 42.0 | 63.3 |
| HAIS [48] [ICCV'21] | 45.7 | 69.9 | 43.5 | 64.1 |
| OccuSeg [54] [CVPR'20] | 48.6 | 67.2 | 44.2 | 60.7 |
| SoftGroup [55] [CVPR'22] | 50.4 | 76.1 | 46.0 | 67.6 |
| SSTNet [46] [ICCV'21] | 50.6 | 69.8 | **49.4** | 64.3 |
| **Ours** | **51.2** | **78.1** | 47.8 | **68.4** |

## 5 Additional Ablation Experiments

**Prototypes number in scenes with different instance numbers.** Our experiments encompassed scenes with varying numbers of instances. We established the number of instance prototypes based on the percentage of training samples for each class. Specifically, in the S3DIS dataset, the training points per class range from 0.2% to 38.8%. We assigned $K = 1$ to classes with 0.2% to 1% training samples, $K = 2$ for 1% to 10% samples, $K = 4$ for 10% to 20% samples, $K = 6$ for 20% to 30% samples, and $K = 10$ for classes with over 30% samples. This approach resulted in slightly better performance than using a fixed $K = 10$ for all classes, as shown in Table 7. However, in our current version, we opted to use $K = 10$ for all classes for simplicity.

**The effect of the prototypes number on efficiency.** Table 8 highlights the relationship between the number of prototypes and the efficiency of the semantic segmentation task on the S3DIS dataset. We found that the number of prototypes doesn't markedly influence the model's computational efficiency.

Table 7: Ablative studies of prototypes number setting on S3DIS [1] Area 5.

| $K$ range | S3DIS mIoU |
|---|---|
| unique value | 72.3 |
| $[1, 10]$ | 72.4 |

Table 8: Ablative studies of prototypes number and efficiency on S3DIS [1] Area 5.

| # Prototype | Training speed | Inference speed |
|---|---|---|
| K = 1 | 16.6 h | 400 ms |
| K = 5 | 16.6 h | 402 ms |
| K = 10 | 16.6 h | 404 ms |
| K = 20 | 16.7 h | 420 ms |
| K = 50 | 16.7 h | 430 ms |

# 6 Qualitative Results on Point Cloud Segmentation

Figs. 1-4 illustrate a few representative visual examples of semantic and instance segmentation results on S3DIS [1] and ScanNet V2 [2] datasets, respectively.

# 7 More systemic insight

**Limitation.** Despite the robustness of our method, one limitation is the necessity of a Sinkhorn-Knopp step during the training process (Eq.10), thereby introducing an the additional time complexity $\widetilde{O}(\frac{n^2}{\epsilon^3})$. It should be noted that in real-world implementation, this procedure contributes only a marginal computational load with a few iterations ($S = 3$ in Algorithm 1), requiring approximately 2.5 milliseconds to segregate 10,000 points into $K = 10$ prototypes.

We have compared our approach's mIoU scores, training speed, and inference speed with the baseline model PTv2 [27] using the S3DIS dataset, as shown in Table 9. The inference speed, averaged over the validation set, was measured on an NVIDIA RTX A100 GPU. Our findings indicate that introducing prototype updates results in only a roughly $\sim 4\%$ delay during training yet leads to significant performance improvements. Notably, while the number of parameters and GFLOPs remain comparable, our method achieves a faster inference speed, mainly because there is no added computational overhead during the inference phase.

**Social Impact.** On the upside, our method shows considerable promise for applications like autonomous vehicles, medical imaging. Nonetheless, it's crucial to recognize potential drawbacks.

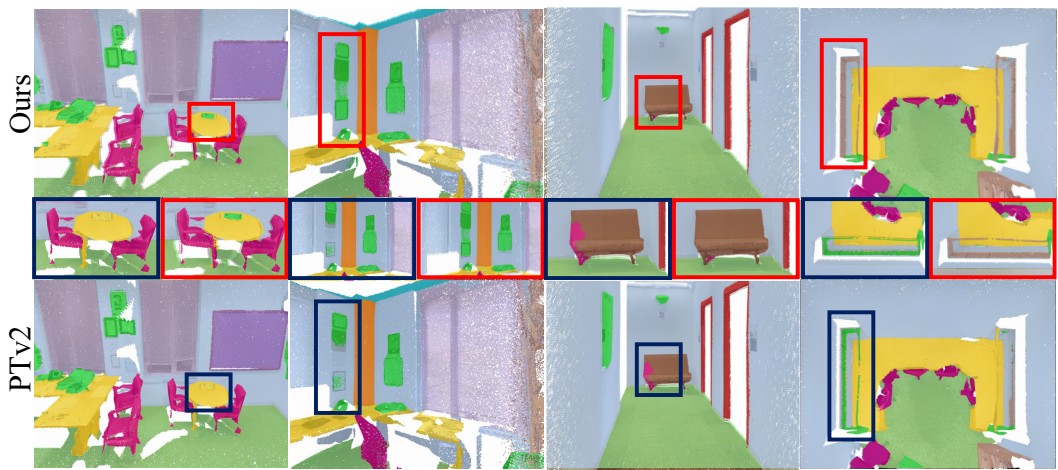

Figure 1: Qualitative semantic segmentation results of PTv2 [27] and PROTOSEG on S3DIS [1] Area 5. Red and blue bounding boxes represent the same zoom-in area on two methods, respectively.

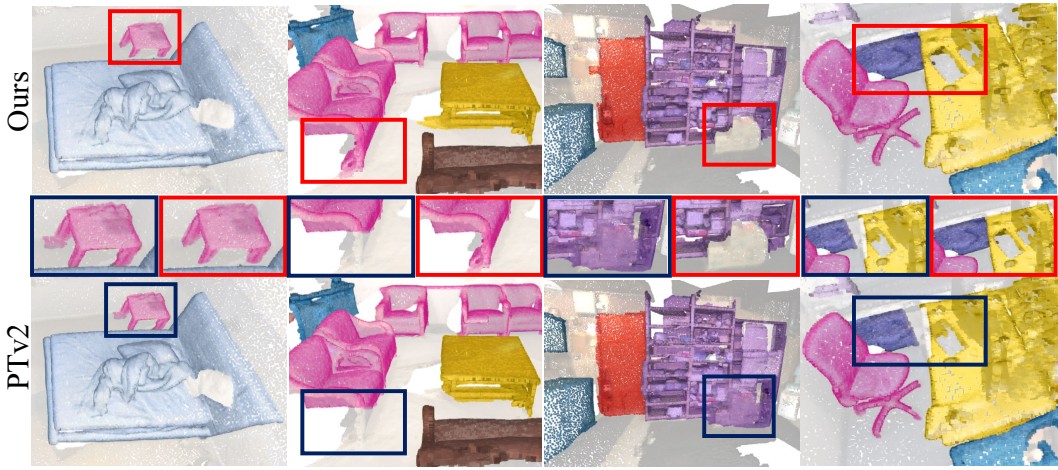

Figure 2: Qualitative semantic segmentation results of PTv2 [27] and PROTOSEG on ScanNet V2 [2] *test* set. Red and blue bounding boxes represent the same zoom-in area on two methods, respectively.

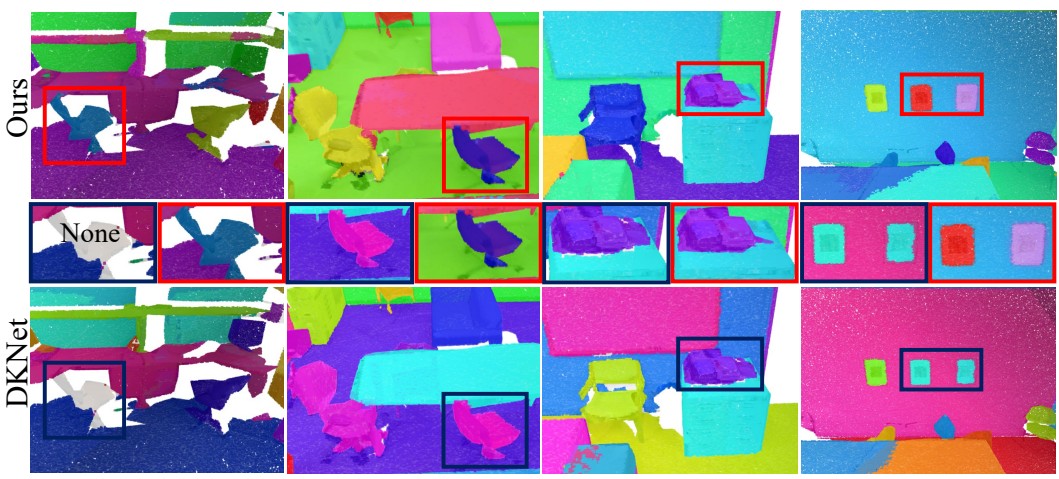

Figure 3: Qualitative instance segmentation results of DKNet [49] and PROTOSEG on S3DIS [1] Area 5. Red and blue bounding boxes represent the same zoom-in area on two methods, respectively.

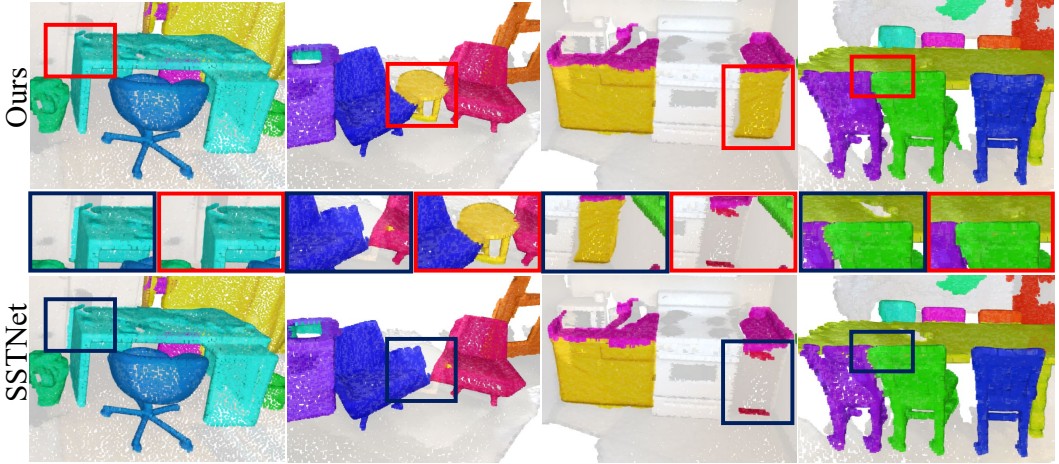

Figure 4: Qualitative instance segmentation results of SSTNet [46] and PROTOSEG on ScanNet V2 [2] *test* set. Red and blue bounding boxes represent the same zoom-in area, respectively.

Table 9: Analysis of computation efficiency on S3DIS [1] Area 5.

| Model | mIoU | # Parameters | GFLOPs | # Epoch | Training speed (hour/epoch) | Inference speed |
|---|---|---|---|---|---|---|
| PTv2 [27] | 71.6 | 3.5m | 6.8 | 100 | 0.160 | 420 ms |
| Our | 72.3 | 3.5m | 6.8 | 100 | 0.166 | 404 ms |

Inaccurate predictions in hands-on settings could risk safety. To mitigate any adverse impact, we recommend implementing a stringent security protocol should our method fail to operate as expected in real-world contexts. We will incorporate this discussion in our revised version.

**Future Work.** Building on above identified limitations, our future work will focus on several key areas. Firstly, we aim to explore more efficient stereotype associations to reduce time complexity, thereby enhancing the overall performance of our model. Additionally, we propose integrating our prototype-anchored classification mechanisms with large-scale models known for their interpretability. This integration is anticipated to augment the accuracy of our algorithm significantly.

To further improve the adaptability, we will investigate the application of our model to more complex open scenarios. These may include tasks such as shape segmentation, detection and tracking. By extending the application of our algorithm, we aim to ensure its utility in a broader range of contexts.

Finally, given the inherent similarity-/distance-based nature of our model, an additional area of focus will be to enhance its interpretability. The goal is to ensure our model's decision-making processes are understandable, increasing its trustworthiness and usability and extending its potential for application in diverse real-world scenarios.

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

**Algorithm 1** Pseudo-code of PROTOSEG in a PyTorch-like style.

```
# P: prototypes (C x K x D)
# E: point embeddings (N x D)

# C: number of classes
# K: number of prototypes for each class
# S: sinhorn-knopp iteration number
# kappa (Eq.9), mu (Eq.12), tau (Eq.13), alphe, beta (Eq.16): hyper-parameters

def ProtoSEG(E, P, label)
    #== Model Prediction and Training Loss (Eq.7) ==#
    # point-to-prototype assignment (N x K x C, Eq.5)
    M = torch.einsum('nd,ckd->nkc', E, P) # permutation matrix (C x K x N)
    output = torch.amax(M, dim=1)

    # training loss
    PPS_loss = -torch.log(F.softmax(M / tau, dim=-1)) # Eq. 13
    PPC_loss = torch.pow(1 - output, 2) # Eq. 14
    PCS_loss = CrossEntropyLoss(output, label) # Eq. 15
    total_loss = PCS_loss + alpha * PPS_loss.mean() + beta * PPC_loss.mean()

    #== Prototype Association (Eq.10) and Update (Eq.12) ==#
    for c in range(C)
        M*_c = prototype_association(M_c, S) # M_c (K x N_c)
        P = prototype_update(E, M, P, M*_c, c)

    return total_loss

def prototype_association(M_c, S)
    M_c = torch.exp(M_c / kappa)
    M_c /= torch.sum(M_c)

    for _ in range(S):
        # normalize each row
        M_c /= torch.sum(M_c, dim=1, keepdim=True)
        M_c /= K
        # normalize each column
        M_c /= torch.sum(M_c, dim=0, keepdim=True)
        M_c /= N_c
    # make sure the sum of each column to be 1
    M_c *= N_c

    return one_hot(M_c)

def prototype_update(E, M, P, M*_c, c)
    # assignments and embeddings for points in class c
    m_c = M[label == c]
    e_c = E[label == c, ...]

    # find points that are assigned to each prototype and correctly classified
    m_c_tile = repeat(m_c, tile=K)
    m_q = M*_c * m_c_tile

    # find points with label c that are correctly classified
    e_c_tile = repeat(m_c, tile=e_c.shape[-1])
    e_c_q = e_c * e_c_tile
    f = torch.mm(m_q.transpose(), e_c_q)

    # num assignments for each prototype of class c
    n = torch.sum(m_q, dim=0)

    # calibration factor
    p_d = torch.mm(e_c_q.transpose(), P[c, n != 0, :])
    w_c = torch.sigmoid(p_d)

    # momentum update (Eq.8)
    if torch.sum(n) > 0:
        P_c = mu * P[c, n != 0, :] + w_c * (1-mu) * f[n != 0, :]
        P[c, n != 0, :] = P_c

    return P
```