# OpenReview forum: "Unified 3D Segmenter As Prototypical Classifiers"
_NeurIPS.cc/2023/Conference — NeurIPS 2023 poster_

### Official Review · Reviewer_Mj9E · 2023-07-03

**Soundness:** 3 good
**Presentation:** 3 good
**Contribution:** 3 good
**Rating:** 5
**Confidence:** 4

**Summary:**

In this paper, the authors propose to use prototype paradigm to develop a generic framework for semantic segmentation, instance segmentation and panoptic segmentation. The proposed ProtoSeg use prototype association to distribute points to different prototypes and dynamic update prototypes according to the assignments. Three optimization terms are introduced to realize different level of distance regularization.

**Strengths:**

1. The proposed prototype-based method is reliable and convincing.
2. The discussed problem of designing unified segmentation model is interesting and important.
3. The performance of the proposed method is strong and achieves SOTA under most conditions.
4. The ablations and discussions are thorough to analyze the designing of the model.
5. The illustrations are clear and helpful for better comprehension.


**Weaknesses:**

1. Limitation discussion is missing.
2. In ablation studies, why use OAcc as metric instead of mIoU? In segmentation, mIoU is more important and more readers care about mioU than OAcc. It would be better if the metric can be switched to mIoU.
3. What is the computation efficiency of the model? Since it is a unified model, will it be much slower than task-specific models? What is the relation between efficiency and prototype number?


**Questions:**

Please refer to the weakness part.

**Limitations:**

There is no explicit limitation discussion section in this paper.

---

> ### Author Rebuttal · Authors · 2023-08-09
>
> We thank reviewer Mj9E for the valuable time and constructive feedback. We provide a point-to-point response below.
> ### Weaknesses 1: Limitation
>
> Sorry for the confusion. We discussed some limitations of this work in the supplementary material, which is related to model efficiency.
>
> ### Weaknesses 2: Metric of the ablation studies
>
> Thanks for your valuable comments. We understand that mIoU (mean Intersection over Union) is a widely recognized and essential metric in segmentation tasks. The following table presents the mIoU scores in the ablation experiments. We will use OAcc and mIOU as indicators of ablation experiments in the revised version.
>
> Ablation studies of Key Components
>
> | Prototypical | $\mathcal{L}_{\text{PCS}}$ | $\mathcal{L}_{\text{PPS}}$ | $\mathcal{L}_{\text{PPC}}$ | mIoU  |
> |--------------|----------------------------|----------------------------|----------------------------|-------|
> | Classifiers  | (Eq.15)                    | (Eq.13)                    | (Eq.14)                    | (\%)  |
> |              | √                     |                            |                            | 71.53 |
> | √       | √                     |                            |                            | 71.74 |
> | √       | √                     | √                     |                            | 72.07 |
> | √       | √                     |                            | √                     | 72.08 |
> | √       | √                     | √                     | √                     | 72.34 |
>
>
> Ablation studies of Prototype Number $K$
>
> | # Prototype | mIoU (%) |
> |--------------|-----------|
> | K=1     | 71.48     |
> | K=5     | 71.84     |
> | K=10    | 72.34     |
> | K=20    | 72.18     |
> | K=50    | 72.05     |
>
> Ablation studies of Distance Measure
>
> | Distance    | mIoU (%)   |
> |-------------|------------|
> | Standard    | 72.10      |
> | Huberized   | 72.17      |
> | Cosine      | 72.34      |
>
> Ablation studies of Momentum Coefficient
>
> | Coefficient $\mu$ | mIoU (%) |
> |-------------------|-----------|
> | 0.9         | 71.62     |
> | 0.99        | 71.88     |
> | 0.999       | 72.34     |
> | 0.9999      | 72.04     |
>
> Ablation studies of Class Balance
>
> | Strategy    | mIoU (%) |
> |-------------|-----------|
> | w/o      | 71.49     |
> | $w^c$   | 72.34     |
> | $M_{p}$ | 72.14     |
> | $M_{n}$ | 71.83     |
>
> Ablation studies of Sinkhorn Iterations $S$
>
> | Iterations (Eq.10) | mIoU (%) |
> |------------|-------|
> | $S=1$    | 71.29 |
> | $S=3$    | 72.34 |
> | $S=5$    | 71.43 |
>
>
> ### Weaknesses 3: Computation efficiency
>
> Thanks for your careful review. (i) Computation Efficiency: We present a comparison between our proposed method and the baseline model, PTv2, in terms of the number of parameters, GFLOPs, and inference speed. The inference speed is averaged over the validation set, and both GFLOPs and inference speed measurements are obtained using an NVIDIA A100 GPU.
>
> | Model | mIoU | \# Parameters | GFLOPs | # Epoch | Training speed (hour/epoch) | Inference speed |
> |-------|------|---------------|--------|--------|--------|-----------------|
> | PTv2  | 71.6 | 3.5m          | 6.8    | 100    | 0.16 h           | 420 ms          |
> | Our   | 72.3 | 3.5m          | 6.8    | 100    | 0.166 h         | 404 ms          |
>
>
> (ii) Inference Speed: One limitation of our algorithm arises from the additional clustering loops (referenced in Eq.10) present in each training iteration. These loops could potentially influence computation efficiency in terms of time complexity. However, the inclusion of three recursive clusterings ensures global model convergence and introduces only a slight computational overhead. This overhead leads to approximately a 4\% reduction in training speed compared to the baseline model PTv2. Notably, while the number of parameters and GFLOPs remain comparable, our method yields better results and a faster inference rate.
>
> (iii) The table below highlights the relationship between the number of prototypes and the efficiency of the semantic segmentation task on the S3DIS dataset. We found that the number of prototypes doesn't markedly influence the model's computational efficiency. We will elaborate on efficiency considerations in our revised version.
>
> | \# Prototype | # Epoch | Training speed (hour/epoch) | Inference speed |
> |--------------|--------|----------------|-----------------|
> | K = 1        |  100    |0.166 h         | 400 ms          |
> | K = 5        |  100    |0.166 h         | 402 ms          |
> | K = 10       |  100    |0.166 h         | 404 ms          |
> | K = 20       |  100    |0.167 h         | 420 ms          |
> | K = 50       |  100    |0.167 h         | 430 ms          |
>
> We appreciate your thoughtful review again and hope we have addressed your concerns. Please let us know if you'd like any further information.

---

### Official Review · Reviewer_saLq · 2023-07-05

**Soundness:** 4 excellent
**Presentation:** 4 excellent
**Contribution:** 3 good
**Rating:** 6
**Confidence:** 4

**Summary:**

This paper introduces PROTOSEG, a prototype-based model that aims to unify semantic, instance, and panoptic segmentation on point cloud. It treats these tasks as a classification problem with varying levels of granularity and achieves high performance on multiple benchmarks.

**Strengths:**

1. The research topic addressed in this paper is the unification of three different segmentation tasks, which holds significant importance.
2. The overall structure of the paper is well-presented and easy to follow.
3. The formulations provided are detailed and comprehensive.
4. The proposed method demonstrates excellent performance on both indoor and outdoor benchmarks.

**Weaknesses:**

1. The analysis of the parameters, flops, and time consumption of the proposed PROTOSEG is missing in this paper. The backbone network used is PTv2, and the original prediction head is replaced with the proposed prototypical classifier. Therefore, it is important to understand how this replacement affects the parameters, flops, and training/testing time, particularly considering that Sinkhorn-Knopp iteration is required to solve equation 9.
2. How can we obtain the prototype instances in Figure 3 from the class sub-centroids, which are D-dimensional vectors?
3. The ablation studies in Table 6 use OAcc as the metric, but mIoU is more commonly used in the Area5 S3DIS. It would be helpful if the authors explain why they chose OAcc and also provide the results of mIoU.
4. The released code of PTv2 on S3DIS differs from the original paper and can achieve a higher mIoU of 72.0+ on Area 5. PROTOSEG achieves a comparable mIoU of 72.3 on S3DIS but notably improves to 76.4 mIoU on ScanNetV2. It would be helpful if the authors could provide some analysis or insights regarding this difference.

**Questions:**

Please refer to the **Weaknesses**.

**Limitations:**

Please refer to the **Weaknesses**.

---

> ### Author Rebuttal · Authors · 2023-08-09
>
> We thank reviewer Mj9E for the valuable time and constructive feedback. We provide point-to-point response below.
>
> ### Weaknesses 1: Parameters, flops, and speed consumption
>
> We detail the complexity analysis in the supplementary material. Although the Sinkhorn-Knopp iteration is utilized, in practice, this procedure adds only a minimal computational burden and affects speed marginally over a few iterations. We present a comparison between our proposed method and the baseline model, PTv2, in terms of the number of parameters, GFLOPs, and inference speed. The inference speed is averaged over the validation set, and both GFLOPs and inference speed measurements are obtained using an NVIDIA A100 GPU.  We will elaborate on efficiency considerations in our revised version.
>
> | Model | mIoU | \# Parameters | GFLOPs | # Epoch | Training speed (hour/epoch) | Inference speed |
> |-------|------|---------------|--------|--------|--------|-----------------|
> | PTv2  | 71.6 | 3.5m          | 6.8    | 100    | 0.16 h           | 420 ms          |
> | Our   | 72.3 | 3.5m          | 6.8    | 100    | 0.166 h         | 404 ms          |
>
> ### Weaknesses 2: Prototype instances based on the class sub-centroids
>
> Sorry for the confusion. To investigate Ad-hoc Explainability, we train the model in the usual manner and calculate the class sub-centroids as prototypes. During the final stages of training, each sub-centroid is anchored to the training points closest to it, determined by the cosine similarity of their embeddings. Specifically, in the last 30 epochs, the prototypes are updated solely based on the embeddings of their corresponding anchored training points. As a result, the prototype instance depicted in Figure 3 represents the partial point cloud that is most similar to the class sub-centroids. The supplementary material provides a more comprehensive experimental setup for the ad-hoc explainability study.
>
> ### Weaknesses 3: Metric of the ablation studies
>
> Thanks for your valuable comments. We acknowledge that mIoU is a commonly used metric in the Area5 S3DIS benchmark. The following table presents the mIoU scores in the ablation experiments. We will use both OAcc and mIOU as indicators of ablation experiments in the revised version.
>
> Ablation studies of Key Components
>
> | Prototypical | $\mathcal{L}_{\text{PCS}}$ | $\mathcal{L}_{\text{PPS}}$ | $\mathcal{L}_{\text{PPC}}$ | S3DIS  | ScanNetv2 |
> |--------------|----------------------------|----------------------------|----------------------------|--------|-----------|
> | Classifiers  | (Eq.15)                    | (Eq.13)                    | (Eq.14)                    | (mIoU) | (mIoU)    |
> |              | √                    |                            |                            | 71.53  | 75.36     |
> | √       | √                     |                            |                            | 71.74  | 75.84     |
> | √       |√                     | √                     |                            | 72.05  | 76.07     |
> | √      | √                     |                            | √                     | 72.06  | 76.12     |
> | √       | √                     | √                     |√                     | 72.34  | 76.32     |
>
>
> Ablation studies of Prototype Number $K$
>
> | # Prototype | S3DIS mIoU (%) |
> |--------------|-----------|
> | K=1     | 71.48     |
> | K=5     | 71.84     |
> | K=10    | 72.34     |
> | K=20    | 72.18     |
> | K=50    | 72.05     |
>
> Ablation studies of Distance Measure
>
> | Distance    | S3DIS mIoU (%)   |
> |-------------|------------|
> | Standard    | 72.10      |
> | Huberized   | 72.17      |
> | Cosine      | 72.34      |
>
> Ablation studies of Momentum Coefficient
>
> | Coefficient $\mu$ | S3DIS mIoU (%) |
> |-------------------|-----------|
> | 0.9         | 71.62     |
> | 0.99        | 71.88     |
> | 0.999       | 72.34     |
> | 0.9999      | 72.04     |
>
> Ablation studies of Class Balance
>
> | Strategy    | S3DIS mIoU (%) |
> |-------------|-----------|
> | w/o      | 71.49     |
> | $w^c$   | 72.34     |
> | $M_{p}$ | 72.14     |
> | $M_{n}$ | 71.83     |
>
> Ablation studies of Sinkhorn Iterations $S$
>
> | Iterations (Eq.10) | S3DIS mIoU (%) |
> |------------|-------|
> | $S=1$    | 71.29 |
> | $S=3$    | 72.34 |
> | $S=5$    | 71.43 |
>
>
> ### Weaknesses 4: Improvement on S3DIS and ScanNetV2 datasets
>
> Good point! Thanks to PROTOSEG's intuitive mechanism, the statistical significance of the class sub-centroid allows it to automatically discern the potential distribution of each class and capture a representative set of local means. In contrast, PTv2, a non-prototype approach, learns only a single weight (or query) vector per class. PROTOSEG is better equipped to adapt to the rich in-class variations, leading to a marked performance improvement on ScanNetV2, a larger dataset with a greater number of classes.
>
> We sincerely appreciate your thorough review and invaluable feedback, which have significantly contributed to the improvement of our work. We hope we have addressed all of your concerns. Please let us know if you require any additional information.

---

> > ### Comment · Reviewer_saLq · 2023-08-13
> >
> > Thank you for your detailed response. Your additional explanations and experimental results have basically addressed my concerns. Please incorporate these contents into the revised version for publication. Additionally, I recommend releasing the source code and pretrained models to benefit the community. I will maintain my positive rating.
> >
> > One minor mistake:
> > >We thank reviewer Mj9E for the valuable time and constructive feedback. We provide point-to-point response below.
> >
> > I am the Reviewer saLq.

---

> > > ### Author Response · Authors · 2023-08-13
> > > **Thanks for your response**
> > >
> > > We extend our sincere gratitude for your valuable response. Your perceptive observations have significantly elevated the quality and lucidity of our manuscript. In light of your guidance, we will seamlessly integrate these supplementary explanations and experimental findings into the revised version. Furthermore, we are unequivocally aligned with your insightful recommendation to make the source code and pretrained models publicly available, thereby fostering an environment conducive to continuous research and advancement within the field. Once again, we are grateful for your meticulous dedication and thorough examination of our work.

---

### Official Review · Reviewer_VQFt · 2023-07-05

**Soundness:** 3 good
**Presentation:** 3 good
**Contribution:** 3 good
**Rating:** 6
**Confidence:** 3

**Summary:**

This work aims to unify different 3D segmentation tasks, i.e., semantic, instance, and panoptic segmentation on point clouds. The authors propose unified prototypical classifiers --- ProtoSeg -- on top of a Transformer backbone, for multi-task 3D segmentation. The main idea is to establish prototype association and update so that the model would be able to output task-specific predictions based on prototypes with different levels of granularity. The authors conducted semantic and instance segmentation tasks on S3DIS and ScanNet V2 and panoptic segmentation on SemanticKITTI. The results show that ProtoSeg outperforms prior works on most evaluation metrics.

**Strengths:**

- S1: This work contributes to the area of multi-task 3D segmentation, which is definitely a valuable research direction for point cloud understanding.

- S2: The proposed ProtoSeg is showing good explainability during the point cloud segmentation process.

- S3: Convincing results on S3DIS and ScanNet V2 demonstrate the effectiveness and superiority.

- S4: Detailed implementation examples, including pseudo-code, are provided in the supplementary material, which is helpful in facilitating future research.


**Weaknesses:**

- W1: Lacking discussions with multi-task image segmentation methods. The pursuit of unifying semantic, instance, and panoptic segmentation on images is not new for the image community. In fact, several works have explored multi-task image segmentation from a mask proposal view (or other related aspects), which is directly linked to the *'prototypical classifier'* in ProtoSeg. The authors are recommended to discuss the main differences between these methods.
  - MaX-DeepLab [R1] proposed a task-conditioned joint training strategy with a task token to support multi-task training and inference.
  - MaskFormer [R2] proposed a simple mask classification model which predicts a set of binary masks, each associated with a single global class label prediction, for semantic- and instance-level segmentation tasks.
  - K-Net [R3] proposed a group of learnable kernels for unified image segmentation, where each kernel is responsible for generating a mask for either a potential instance or a stuff class.
  - Mask2Former [R4] and OneFormer [R5] extended MaskFormer [R2] for better segmentation accuracy.
  - kMaX-DeepLab [R6] proposed kMaX decoders that transform cluster centers into mask embedding vectors, which multiply with the pixel features to generate the predicted masks.

- W2: It is a bit strange that the authors conducted semantic segmentation experiments on S3DIS and ScanNet V2, but not SemanticKITTI. Although the panoptic segmentation results of ProtoSeg (Table 5) seem reasonable, the authors are recommended to supplement the semantic segmentation results as well to further consolidate the effectiveness.

- Minor: The training complexity of ProtoSeg seems high, since it introduces extra losses for optimization and extra procedures (e.g. prototype update). Although it is not a major concern in this work, the authors are recommended to compare the complexity of ProtoSeg with the baseline models.

#### References
[R1] H. Wang, et al. "Max-DeepLab: End-to-End Panoptic Segmentation with Mask Transformers," CVPR, 2021.

[R2] B. Cheng, et al. "Per-Pixel Classification is Not All You Need for Semantic Segmentation," NeurIPS, 2021.

[R3] W. Zhang, et al. "K-Net: Towards Unified Image Segmentation," NeurIPS, 2021.

[R4] B. Cheng, et al. "Masked-Attention Mask Transformer for Universal Image Segmentation," CVPR, 2022.

[R5] J. Jain, et al. "OneFormer: One Transformer to Rule Universal Image Segmentation," CVPR, 2023.

[R6] Q. Yu, et al. "K-Means Mask Transformer," ECCV, 2022.


**Questions:**

- Q1: Is the *'IF-Then'* explanation logic applicable to outdoor point clouds? For example, can it also interpret the panoptic segmentation process on SemanticKITTI?

- Q2: How to ensure the convergence of $u^c$ and $v^c$ in the Sinkhorn-Knopp iteration?

- Q3: Ablation study, do you have some analysis or observations on the ScanNet V2 dataset? The scores on the S3DIS benchmark seem saturated and hard to depict influencing factors.

- Minor 1: Line 33, what is the meaning of *'epistemological'* here?

- Minor 2: Line 109, *'dynamic'* should be revised to 'dynamically'.



**Limitations:**

The authors discussed some limitations of this work in the supplementary material, which is related to model efficiency. The authors did not discuss the potential negative societal impact of their work.

---

> ### Author Rebuttal · Authors · 2023-08-09
>
> Thanks reviewer VQFt for the valuable time and constructive feedback. We provide point-to-point response below.
> ### W1: Discussions with multi-task image segmentation methods
> Thank you for your reminder. As you mentioned, we plan to introduce a new subsection in Section 4 to address multi-task image segmentation, which is intimately related to our work.
>
> Multi-task Image Segmentation aims to develop a cohesive architecture to address various segmentation challenges. K-Net [R3] pioneered this approach, leveraging dynamic kernel learning for mask generation. The dynamic kernel updating mechanism and binary matching strategy within K-Net underline its versatility and straightforwardness. Of note, this method stands out due to its ability to significantly reduce memory usage and improve inference speeds, making it especially suitable for real-time applications.
>
> In recent developments, several architectures inspired by DETR have formulated different tasks within a mask classification paradigm, utilizing a transformer decoder with object queries. For instance, Max-DeepLab [R1] negates the need for box predictions by employing conditional convolutions, bridging the gap between box-dependent and box-independent methods for the first time. kMaX-DeepLab [R6], which views the object queries as cluster centers with adaptable embedding vectors, reimagines the cross-attention through the lens of k-means clustering. MaskFormer [R2] and its subsequent versions [R4] [R5] introduce a streamlined and effective inference strategy to merge outputs into a task-specific prediction format using a collection of binary masks.
>
> Unlike the above methods, our approach perceives these three closely-related tasks as a prototype-anchored classification challenge with distinct granularity levels. We define prototypes as class sub-centroids derived from the feature embeddings of training samples. Then, a test sample is directly classified based on its proximity to the nearest centroids. Such an approach, rooted in case-based reasoning, introduces a distinct element of ad-hoc interpretability to our method, as detailed in Section 5.4, Fig. 3.
> ### W2: Semantic seg results for SemanticKITTI
> Thanks. We agree to add SemanticKITTI experiments to support our claim better. The below results show that our method is +0.14% higher in mIoU than current works. We will include more comparisons in the revised version.
>
> | Method | mIoU |
> |---|---|
> | RPVNet [ICCV21]| 70.3 |
> | $(AF)^2$-S3Net [CVPR21]| 70.8 |
> | PVKD [CVPR22]| 71.2 |
> | Ours| **72.6** |
> ### Weaknesses Minor: Complexity
> Thanks. The discussion regarding the complexity of our ProtoSeg can be found in Appendix. In real-world implementations, it's minimal computation. After a few iterations, about 2.5ms to classify 10K points into $K=10$ prototypes. We've compared our method's mIoU scores, training and inference speed with the baseline model PTv2 on S3DIS, as shown in response to Mj9E's W1 due to the word limit. We find that prototype updates cause $\sim4$% delay in training yet improve performance. Our method has faster inference due to no added overhead. These results and analysis would be updated in revised version.
> ### Q1: 'IF-Then' for outdoor point clouds
> Sorry for this confusion. 'IF-Then'[R1] explanation logic, please see Eqs.1-2 in Appendix. Please note that the interpretability of a classifier mainly comes from its decision-making mode, i.e., a test sample is directly classified into the class with the closest centroids instead of the training strategy or datasets. Therefore, 'IF-Then' applies to SemanticKITTI. We will update the related discussion to clarify this point in the revised version.
>
> [R1] Towards explainable deep neural networks (xdnn).
> ### Q2: Convergence of $u^c$ and $v^c$
> Thanks. The Sinkhorn-Knopp iteration involves a regularization parameter $\kappa$ (see Eq.10) that helps balance convergence speed and stability [34]. A smaller $\kappa$ can lead to slower convergence but more accurate results, while a larger $\kappa$ does the opposite. We just use $\kappa=0.05$ following [34] for our experiments, not extensively fine-tuned. We will update the related discussion to clarify this point in the revised version.
> ### Q3: Ablation study on ScanNetV2
> Thanks for your valuable comments. We performed ablation experiments for key components (like Table 6 (a)) on the Scannetv2 dataset to verify the validity of our Prototypical Classifier (Section 3.2) and supervision scheme (Section 3.3). The results are included in the response to saLq'W3 due to word limit. Other parameters are set experientially. To make our conclusions more convincing, we will use both S3DIS and ScanNetv2 datasets for evaluation in the revision.
> ### Questions Minor 1: Meaning of 'epistemological'
> We clarify that the 'epistemological framework' discusses the fundamental conceptual framework that guides our methodology design. Given the irregular and sparse nature of point clouds, we propose that point cloud segmentation naturally aligns with a classification problem. This foundational perspective shapes our research direction and motivates the creation of our prototype-anchored classification method, allowing us to seamlessly unify various segmentation tasks within a single framework.
> ### Questions Minor 2: Some mistaking in Line 109
> Our apologies. Proofreading will be performed in the revised version.
> ### Limitations: Potential negative societal impact
> Thanks. On the upside, our method shows considerable promise for applications like autonomous vehicles, medical imaging. Nonetheless, it's crucial to recognize potential drawbacks. Inaccurate predictions in hands-on settings could risk safety. To mitigate any adverse impact, we recommend implementing a stringent security protocol should our method fail to operate as expected in real-world contexts. We will incorporate this discussion in our revised version.
>
> Thanks for your thorough review. Let us know if you need more information.

---

> > ### Comment · Reviewer_VQFt · 2023-08-18
> >
> > Thank the authors for providing a detailed response.
> >
> > I have read the authors' responses as well as other reviewers' comments. Several reviewers mentioned the model complexity and training overhead, which are indicators that the ProtoSeg framework should pay attention to, especially on large-scale point clouds.
> >
> > What is more, it is a bit supersizing that ProtoSeg can achieve 72.6 mIoU on SemanticKITTI (better than most of the models that are tailored for LiDAR semantic segmentation), considering that it only achieved a 62.4 PQ for the panoptic segmentation task.
> >
> > I believe that the following two pieces of evidence should be important for a more comprehensive evaluation:
> > - Similar to the response to Reviewer saLq, the authors are recommended to conduct a  complexity analysis on SemanticKITTI.
> > - More implementation and configuration details on SemanticKITTI segmentation are needed, such as the performance on the validation set.
> >
> > I will listen to other reviewers' opinions before recommending the final rating.

---

> > > ### Author Response · Authors · 2023-08-19
> > > **Thanks for your response and further discussion**
> > >
> > > Thank you so much for your response.
> > >
> > > We clarify that PQ is used for provides a more comprehensive assessment of segmentation quality by incorporating instance-level information. It rewards correct matches of instances and penalizes over-segmentation and under-segmentation errors. However, mIoU is weighted according to the size of each class in the prediction, which results in a higher mIoU score even if instance-level performance is unsatisfactory. In the point cloud panoptic segmentation task on SemanticKITTI, mIoU is always higher than PQ. As shown in the following table,  GP-S3Net [29] achieved a 60.0 PQ score (2.4 lower than ours), but achieved a mIoU of 70.8 in panoptic segmentation, also better than most of the models that are tailored for LiDAR semantic segmentation. We will report the mIoU scores in the revised version of Table 5.
> > >
> > > |Mothod|  PQ| mIoU|
> > > |---|---|---|
> > > |LPSAD [75]|38.0|50.9 |
> > > |Panoster [76]|52.7|59.9 |
> > > |Panoptic-PolarNet [26]|54.1|59.5|
> > > |DS-Net [51]| 55.9| 61.6 |
> > > |EfficientLPS [50] | 57.4|61.4 |
> > > |GP-S3Net [29] |60.0| **70.8**|
> > > |SCAN[32] |61.5| 67.7|
> > > |Panoptic-PHNet [15] | 61.5| 66.0|
> > > |Our| 62.4| 67.5|
> > >
> > > As you mentioned, we agree to add more details to support our claim about model complexity.
> > > The following tables show the complexity analysis, and the performance comparisons of our model on SemanticKITTI validation set. We achieved a PQ score of 66.3 and a mIou of 75.4, confirming the performance benefits of our model. We will include these results in the revised version.
> > >
> > > |Model  |# Parameters | GFLOPs| # Epoch | Training speed (hour/epoch) | FPS |
> > > |---|---|---|---|---|---|
> > > |Ours|3.5m|6.8|80|~0.30 h|12.1|
> > >
> > > | Method | PQ | PQ$^{\dagger}$ | RQ | SQ | PQ$^{Th}$ | RQ$^{Th}$ | SQ$^{Th}$ | PQ$^{St}$ | RQ$^{St}$  | SQ$^{St}$ | mIoU |
> > > |---|---|---|---|---|---|---|---|---|---|---|---|
> > > | LPSAD [75] | 37.4 | 44.2 | 47.8 | 66.9 | 25.3 | 32.4 | 65.2 | 46.2 | 58.9 | 68.2 | 49.4 |
> > > | Panoster [76] | 55.6 | -    | 66.8 | 79.9 | 56.6 | 65.8 | -    | -    | -    | -    | 61.1 |
> > > | DS-Net [51] | 57.7 | 63.4 | 68.0 | 77.6 | 61.8 | 68.8 | 78.2 | 54.8 | 67.3 | 77.1 | 63.5 |
> > > |Panoptic-PolarNet [26]| 59.1 | 64.1 | 70.2 | 78.3 | 65.7 | 74.7 | 87.4 | 54.3 | 66.9 | 71.6 | 64.5|
> > > | EfficientLPS [50] | 59.2 | 65.1 | 69.8 | 75.0 | 58.0 | 68.2 | 78.0 | 60.9 | 71.0 | 72.8 | 64.9 |
> > > | GP-S3Net [29] | 63.3 | 71.5 | 75.9 | 81.4 | 70.2 | 80.1 | 86.2 | 58.3 | 72.9 | 77.9 | 73.0 |
> > > |Ours| 66.3 | 75.2 | 78.9 | 84.7 | 73.8 | 76.5 | 80.1 |59.6 | 74.8 | 78.5| 75.4|
> > >
> > > Thanks again for your careful review. Please let us know if any other concerns.

---

> > > > ### Comment · Reviewer_VQFt · 2023-08-19
> > > >
> > > > Many thanks for the follow-up clarification.
> > > >
> > > > I have the remaining questions:
> > > > - Can you provide the results on the validation set of SemanticKITTI for *semantic segmentation*?
> > > > - Can you provide the results of your baseline model as well? I believe it should be a variant of PTv2.
> > > > - PTv2 was designed for indoor point cloud understanding. It is supersizing that the proposed model (which uses a backbone from PTv2) can achieve such promising results on SemanticKITTI. It is well-observed that a Transformer-based backbone will require significantly more computation resources on large-scale outdoor point clouds. Thus, it would be helpful for further clarification if the authors can provide more evidence in this aspect, for example: 1) complexity comparison with existing outdoor point cloud segmentation models, 2) more details on the model implementation, training and testing configurations, etc.

---

> > > > > ### Author Response · Authors · 2023-08-21
> > > > > **Appreciating your reply and extending the conversation**
> > > > >
> > > > > We appreciate your interest in our work and the opportunity to clarify our research further.
> > > > >
> > > > > ### Results of our model and baseline
> > > > > Regarding the results of the validation set of SemanticKITTI for semantic segmentation, we are pleased to provide the results of our model and baseline (i.e., a variant of PTv2) in the following table. We will include these results in the revised version.
> > > > >
> > > > > |Mothod| mIoU|
> > > > > |---|---|
> > > > > |Cylinder3D [CVPR21] | 65.9|
> > > > > |2DPASS [ECCV22]| 69.3|
> > > > > |(AF)2-S3Net [CVPR21] | 74.2|
> > > > > |Our Baseline | 70.3|
> > > > > | Ours| 74.2|
> > > > >
> > > > > ### Complexity comparison
> > > > > As you noticed, Transformer-based backbones tend to demand more computational power when applied to large-scale outdoor point clouds. As shown in the following table, compared with existing methods, our model has larger parameters and GFLPs, but thanks to the intuitive working mechanism of our model, our model demonstrates a higher mIoU while also achieving commendable FPS. This resonates with the crux of our ongoing research focus—reducing the computational burden without compromising segmentation efficacy. We will include a detailed complexity comparison with existing outdoor point cloud segmentation models in our revised version and discuss the future of work.
> > > > >
> > > > > |Method|mIoU|# Parameters (M)|GFLOPs| FPS|
> > > > > |---|---|---|---|---|
> > > > > |RPVNet [ICCV21]|70.3|1.5| 6.7|14.8|
> > > > > |(AF)2-S3Net [CVPR21]|70.8| 1.8 | 7.2 |11.8|
> > > > > |PVKD [CVPR22]|71.2|0.9|3.6|20.0|
> > > > > |Ours|**72.6**|3.5|6.8|12.1|
> > > > >
> > > > > ### More details
> > > > > Furthermore, we understand the importance of transparency in model implementation, training, and testing configurations. In the revised version, we will add the model parameter settings on the SemanticKITTI semantic segmentation task as follows. We will provide comprehensive details in the code resources to be released.
> > > > >
> > > > > |Lr | Optimizer | Decay | K | µ |τ |κ |α |β |
> > > > > |---|---|---|---|---|---|---|---|---|
> > > > > |$1\times 10^{-3}$ |AdamW|0.2| 8 |0.999|0.1|0.05|0.01|0.01|
> > > > >
> > > > > Once again, we sincerely value your invaluable feedback and remain receptive to any further inquiries you may have.

---

> > > > > > ### Comment · Reviewer_VQFt · 2023-08-21
> > > > > >
> > > > > > Thanks to the authors for the follow-up response.
> > > > > >
> > > > > > Since the authors have provided more details on the model configuration and evaluations, I have no further questions regarding the effectiveness of the proposed approach.
> > > > > >
> > > > > > I suggest the authors include the following aspects in the revised paper:
> > > > > > - Comparisons of model complexity, as also required by Reviewers saLq and Mj9E;
> > > > > > - Results on both the validation and testing sets of SemanticKITTI;
> > > > > > - The 'IF-Then' explanation logic on SemanticKITTI;
> > > > > > - Discussion with multi-task image segmentation works;
> > > > > > - Discussion of more recent point cloud semantic and panoptic segmentation works;
> > > > > > - Correction of typos and misunderstandings;
> > > > > > - Other factors required by the other three reviewers.
> > > > > >
> > > > > > I believe this work is of good value. I will keep my rating as "Weak Accept". I encourage the authors to open-source this project so that follow-up research can use their designs.

---

> > > > > > > ### Author Response · Authors · 2023-08-21
> > > > > > > **Review acknowledgment**
> > > > > > >
> > > > > > > We sincerely appreciate your thoughtful and comprehensive feedback on our paper. We are pleased that the additional information regarding the results and model configuration has effectively addressed your concerns. We acknowledge your recommendations for enhancing the revised paper, and we are committed to integrating these aspects to fortify the value of our work further. We concur with your perspective that open sourcing can catalyze collaboration and progress in the field. Once again, we extend our gratitude for your time and invaluable insights.

---

### Official Review · Reviewer_MoHU · 2023-07-06

**Soundness:** 3 good
**Presentation:** 2 fair
**Contribution:** 3 good
**Rating:** 4
**Confidence:** 4

**Summary:**

The paper presents a unified 3D segmenter to unify different segmentation tasks, including semantic, instance, and panoptic segmentation, for 3D point clouds.  It proposes to define the segmentation task as prototype classification and designs the prototype association and prototype update process.

**Strengths:**

1. The paper aims to unify different 3D segmentation models for multiple segmentation tasks, which is a problem worth exploring.
2. The ProtoSeg method achieves excellent performance on multiple tasks (semantic, instance, panoptic segmentation), showing better results than previous methods in multiple datasets (S3DIS, ScanNet, SemanticKITTI).

**Weaknesses:**

1. In Section 2, the paper introduces different models for different 3D segmentation tasks. However, the difficulty of unifying these tasks is not clearly stated. I expect to see the challenges of unifying segmentation tasks in Section 2 and the corresponding methods for tackling the challenges in Section 3.
2. More figures are expected to illustrate the prototype association and prototype update process.
3. I'd like to see how the prototype classifier is used to unify the different 3D segmentation tasks, especially for the instance and panoptic segmentation tasks, but it is only very simply introduced in Section 3.4 without any details.
4. It is not clear how to decide the number of instance prototypes and how to deal with scenes with different instance numbers.

**Questions:**

I like the motivation of the paper and the results of the paper are inspiring. However, the overall paper is not very clear and hard to follow. Please refer to the weaknesses part for the questions.

**Limitations:**

Limitations are discussed in the supplement.

---

> ### Author Rebuttal · Authors · 2023-08-09
>
> We thank reviewer MoHU for the valuable time and constructive feedback. We provide a point-to-point response below.
> ### Weaknesses 1: Challenges and the corresponding methods of unifying segmentation tasks
>
> Thank you for your insightful review. We greatly value your observation and will revise Section 2 to more prominently highlight the challenges associated with unifying diverse 3D segmentation tasks. A key challenge is the inherent customization of segmentation methods for specific tasks. From a broader perspective, it's evident that semantic segmentation methods might not directly apply to instance segmentation tasks and vice versa. For instance, the semantic segmentation method PTv2 [2] [NeurIPS’22] lacks modules for instance-specific localization and segmentation. This makes it inadequate for capturing the detailed spatial relationships required for precise instance boundary delineation and separation. In contrast, the instance segmentation method SoftGroup [47] [CVPR’22] uses grouping mechanisms to link points belonging to the same instance. While such a mechanism is essential for instance segmentation, it could introduce unwarranted complexity and overhead in semantic segmentation scenarios, where distinguishing individual instances isn't the primary goal. On a finer scale, we observe inconsistencies across data representation, feature extraction, and prediction heads (Eqs.1-3) in these methods. To illustrate, let's consider the prediction head: both MLP-based and Query-based methods rely on a single class-specific vector for each class (Eq.3). This approach often falls short in capturing the rich intra-class variance. Moreover, these class-specific vectors are learned fully parametrically without assessing their representational effectiveness.
>
> To address the challenges, we propose a unified framework that interprets different segmentation tasks as unique granularity classification problems. This approach crafts a prototype classifier, eliminating extra computational load and task-specific architectures. This method offers versatility, providing crucial insights into segmentation model design and training strategies. In terms of model design, we unveil a pioneering prototype-anchored classification method. It distinctly defines sub-class centers within the point embedding space as prototypes, eliminating the need for class-specific vectors. During training, prototypes capture dataset features, inherently integrating biases like intra-class compactness and inter-class separation into supplementary optimization criteria. This shapes the embedding space holistically, beyond prediction accuracy optimization. Importantly, our unique model design and training approach grant the advantage of ad-hoc interpretability, enhancing the model's transparency and understandability.
>
> We will include a more detailed description in the revised version for better clarity.
> ### Weaknesses 2: More figures
> Thanks. This suggestion is highly valuable. We will certainly incorporate more illustrations in the revision to explain the details of the prototype assignment and prototype update.
> ### Weaknesses 3: About prototype classifier for instance and panoptic segmentation tasks
> Sorry for not explaining it clearly. For these tasks, we additionally learn fine-grained prototypes $P^{c,o}\in R^{D\times T},P^{c} \in R^{T\times K}$, where $p_t^{c,o}$ denotes the center of the t-th sub-cluster of training point samples belonging to instance $o$ of class $c$. Similarity to Eq.5, we make predictions by comparing point embedding $e_i \in R^D$, with prototype $p^{c,o}_{t}$ and assigning the corresponding prototype's class as the response:
>
> $$p(c,o|x_i)=\frac{\exp(-d_{e_i,c,o})}{\sum^{C,O}\_{c'=1,o'=1}\exp(-d_{e_i,c',o'})}\,\quad with \quad d_{e\_i,c,o} = \arg \min \{ <e_i, p\_t^{c,o}> \}_{t=1}^T.$$
>
> To get the informative prototype, we learn a fine-grained permutation matrix $M$ for prototype assignment and update. Similarity to Eq.6, we aim to maximize the similarities $J(M)$:
>
> $$\max_{M}\mathcal{J}(M)=\text{Tr}((M^{c,o})^{\top}(P^{c,o})^{\top}E^c)$$
>
> where $M^{c,o}$ is a permutation matrix, $m_{t,i}^{c,o} \in M^{c,o}$
> denotes the one-hot assignment vector that assigns the point sample $x_i$ to the prototype $t$ of instance $o$ of class $c$. Prototype association and update are as in Section 3.2. For instance-level supervision, we include cross-entropy loss (Section 3.3).  We will add the detail of this part to the revised version.
> ### Weaknesses 4: Number of instance prototypes
> Sorry for this confusion. We determined the number of instance prototypes needed through ablation experiments in Table 6 (b), Section 5.4. It would be interesting to find ways to ascertain this number automatically. In our exploration, we encountered clustering techniques like [R1] that might help, but complex algorithms led to time challenges. We will elaborate on this in the appendix of our revised version.
>
> Our experiments encompassed scenes with varying numbers of instances. We established the number of instance prototypes based on the percentage of training samples for each class. S3DIS training points per class range from 0.2% to over 30%. We assigned $K=1$ to classes with 0.2-1% training samples, $K=2$ for 1-10% samples, $K=4$ for 10-20% samples, $K=6$ for 20-30% samples, and $K=10$ for classes with over 30\% samples. This approach resulted in slightly better performance than using a fixed $K=10$ for all classes. However, in our current version, we used $K=10$ for all classes for simplicity. We will discuss the number of prototypes and future work in the revised version.
>
> | $K$ range    | S3DIS mIoU |
> |--------------|------------|
> | unique value | 72.3       |
> | $[1,10]$     | 72.4       |
>
> [R1] DeepDPM: Deep Clustering With an Unknown Number of Clusters
>
> Finally, thanks for your valuable comments, which helped position our work relative to previous efforts.

---

### Author Rebuttal · Authors · 2023-08-09

Thank you so much for your careful review and suggestive comments. We will revise our paper according to your comments. The major changes are as follows:

+ We will detail the challenges and methods of unified segmentation tasks, introduce a diagram of the prototyping process, and detail the use of prototype classifiers in instance segmentation and panoramic segmentation tasks, according to Reviewer MoHU's comments. In addition, we will add ablation experiments on the effect of the number of instances on the prototype, and look forward to future work.

+ We will discuss relations to multi-task methods, adds experiments with SemanticKITTI data, thoroughly examines computation efficiency and convergence in Sinkhorn-Knopp iteration, and conducts ablative experiments on Scannetv2, according to Reviewer VQFt's comments. It also ensures language accuracy and discusses societal implications.

+ We will offer more detailed discussions regarding the computation efficiency, including complexity, parameters, flops, and time consumption, according to Reviewer VQFt, saLq, and Mj9E comments.

+ Lastly, as per Reviewer saLq and Mj9E's suggestions, ablative experiment results for mIoU on S3DIS and Scannetv2 will be included.

Sincerely yours,

Authors.

---

### Author Response · Authors · 2023-08-21

Dear AC,

We sincerely appreciate the tremendous efforts of all contributors to the conference.  At the end of the rebuttal process, we want to summarize our endeavor of the submitted work. A prevailing trend in the community is the establishment of unified models capable of managing multiple tasks. Our research represents an early effort in the point cloud domain to develop such a unified model that can address a spectrum of heterogeneous tasks. With the validated empirical evidence, we believe our work can present valuable insights to the community. In addition, we attempted to engage in further discussions with the reviewer MoHU but received no response. The reviewer provided conclusive statements without detailed critiques. Thus, we respectfully disagree. In sum, we are pleased to receive the majority of approval from the reviewers and truly appreciate all their insightful feedback.

Sincerely,

The Authors

---

### Decision · Program_Chairs · 2023-09-21

**Decision:**

Accept (poster)

**Comment:**

The paper proposes a transformer based unified framework for semantic, instance, and panoptic segmentation. Experiments on benchmark datasets confirm the efficacy of the design. Reviewers are initially less clear about the challenge of such a unification, how it is different from multi-task segmentation, complexity of the proposed method, and other technical details; they also require additional experiments on additional benchmarks. The authors provide detailed responses that address these concerns. Please include these additional results in the final version of the paper. Congratulations!